# Reactivation of a developmental *Bmp2* signaling center is required for therapeutic control of the murine periosteal niche

Valerie S Salazar[1,2†], Luciane P Capelo[1,3†], Claudio Cantù[2,4], Dario Zimmerli[2], Nehal Gosalia[5], Steven Pregizer[1], Karen Cox[1], Satoshi Ohte[1,6], Marina Feigenson[1], Laura Gamer[1], Jeffry S Nyman[7], David J Carey[8], Aris Economides[5], Konrad Basler, Vicki Rosen[1]*

[1]Department of Developmental Biology, Harvard School of Dental Medicine, Boston, United States; [2]Institute for Molecular Life Sciences, University of Zürich, Zürich, Switzerland; [3]Instituto de Ciência e Tecnologia, Universidade Federal de São Paulo, São Paulo, Brazil; [4]Wallenberg Centre for Molecular Medicine, Department of Clinical and Experimental Medicine (IKE), Faculty of Health Sciences, Linköping University, Linköping, Sweden; [5]Regeneron Pharmaceuticals, Tarrytown, United States; [6]Department of Microbial Chemistry, Graduate School of Pharmaceutical Sciences, Kitasato University, Tokyo, Japan; [7]Department of Orthopaedic Surgery and Rehabilitation, Vanderbilt University Medical Center, Nashville, United States; [8]Geisinger Health System, Danville, United States

**\*For correspondence:**
vicki_rosen@hsdm.harvard.edu

[†]These authors contributed equally to this work

**Abstract** Two decades after signals controlling bone length were discovered, the endogenous ligands determining bone width remain unknown. We show that postnatal establishment of normal bone width in mice, as mediated by bone-forming activity of the periosteum, requires BMP signaling at the innermost layer of the periosteal niche. This developmental signaling center becomes quiescent during adult life. Its reactivation however, is necessary for periosteal growth, enhanced bone strength, and accelerated fracture repair in response to bone-anabolic therapies used in clinical orthopedic settings. Although many BMPs are expressed in bone, periosteal BMP signaling and bone formation require only *Bmp2* in the *Prx1-Cre* lineage. Mechanistically, BMP2 functions downstream of Lrp5/6 pathway to activate a conserved regulatory element upstream of *Sp7* via recruitment of Smad1 and Grhl3. Consistent with our findings, human variants of *BMP2* and *GRHL3* are associated with increased risk of fractures.
DOI: https://doi.org/10.7554/eLife.42386.001

## Introduction

After birth, the skeleton sustains a period of exuberant longitudinal and periosteal growth, the respective processes by which bones grow in length and width (*Salazar et al., 2016a*). Since increased length and increased width have opposite effects on susceptibility to fracture, longitudinal and periosteal growth must be tightly coupled to build a skeleton that supports body weight and motion during postnatal life (*Pathria et al., 2016*; *Davison et al., 2006*). Longitudinal growth is mediated by growth plate cartilage. Defects in longitudinal growth are well documented in the dwarfism resulting from genetic mutations of FGFR3 (MIM100800) (*Ornitz and Legeai-Mallet, 2017*), PTHR1 (MIM156400) (*Schipani and Provot, 2003*), or GDF5 (MIM200700) (*Salazar et al.,*

*2016a*). Periosteal growth is mediated by the periosteum. Although excessive periosteal bone formation is linked to activating mutations of the canonical WNT pathway (*Baron and Kneissel, 2013*), accounts of defective periosteal growth in humans are extremely rare (*Bonafe et al., 2015*). The identity of the essential signal governing periosteal activity during early postnatal life therefore remains unknown. It also remains unclear whether functions of the adult periosteum, including fracture repair and cortical expansion in response to therapeutic agents, rely on reactivation of this putative developmental signal.

The periosteum is a stratified fibro-cellular structure that adds tissue to the exterior surfaces of bone, much like rings on the trunk of an actively growing tree (*Dwek, 2010*). The outer periosteum provides a collagen-rich barrier between muscle and the underlying bone and is the site of insertion for tendons and ligaments. It is highly vascularized, innervated, and sparsely populated by fibroblasts. The cambium, or inner periosteal layer adjacent to bone, is a repository for self-renewing skeletal progenitors that differentiate into bone-forming osteoblasts (*Dwek, 2010*). The periosteum is thick and highly active during pre-pubertal growth but becomes thin and largely quiescent in adult life. The molecular mechanism controlling the conversion between active and quiescent periosteal states is not well understood.

Since *Bmp2* is essential for initiation of fracture repair (*Tsuji et al., 2006*), we hypothesized that *Bmp2* governs all major developmental and inducible functions of the periosteal niche. To test this, we performed skeletal phenotype analysis of mice where *Bmp2* was selectively ablated in progenitor, committed, or mature osteoblast populations. We mapped the endogenous *Bmp2* expression domain and compared this to the BMP signaling domain during skeletal development and homeostasis. Periosteal growth and fracture phenotypes of *Bmp2* mutant mice were monitored following genetic or pharmacologic activation of the LRP5/6 signaling pathway. We investigated recruitment of pathway-specific transcription factors to genome-wide cis-regulatory elements, establishing at the molecular level the epistatic relationship between canonical WNT and BMP2 signaling during osteoblast differentiation. And finally, we performed phenome wide analysis to test links between our preclinical data and fracture risk in clinical settings.

## Results

### Osteoprogenitor-derived BMP2 couples longitudinal to periosteal bone growth

Removal of *Bmp2* from the developing mouse limb (*Bmp2^{Flox/Flox}; Prx1-Cre*) causes spontaneous fractures that do not heal (*Tsuji et al., 2006*). Fracture repair and bone graft healing were rescued in *Bmp2*-deficient bones by provision of recombinant BMP2 (*Chappuis et al., 2012*), however the underlying cause of the spontaneous fractures remained unknown. During skeletal phenotyping, microcomputed tomography (microCT) revealed that *Bmp2^{Flox/Flox}* (WT) femurs (*Figure 1a*) and *Bmp2^{Flox/Flox}; Prx1-Cre* (*Bmp2* Prx1-cKO) femurs (*Figure 1b*) were indistinguishable at birth. *Bmp2* Prx1-cKO femurs developed a striking geometry after birth, characterized by near normal length (*Figure 1c*) but narrow width (*Figure 1d*). In the radius/ulna, defective periosteal bone growth was not evident at birth (*Figure 1e–f*), but appeared by 2 weeks of age (*Figure 1g–h*) and remained unresolved during adult life. The radius/ulna of WT and *Bmp2* Prx1-cKO mice contained similar proportions of cortical bone and medullary space at birth (*Figure 1i*). By 2 weeks, forelimb structures of *Bmp2* Prx1-cKO mice were composed primarily of cortical bone (*Figure 1j*) despite the total cross-sectional area being dramatically reduced compared to controls. This slender bone phenotype was not restricted to the radius/ulna (*Figure 1g*) and femur (*Figure 1k*) but appeared at all appendicular skeletal sites including the tibia (*Figure 1l*) and metatarsals (*Figure 1m*). Osteopenia was not evident in the axial skeleton where *Prx1-Cre* is not active (*Durland et al., 2008*; *Logan et al., 2002*).

Defective periosteal bone growth occurred in males and females and was therefore unlikely to be caused by sex hormones. IGF-1 signaling can affect bone width (*Lindsey and Mohan, 2016*), but *Bmp2* Prx1-cKO mice expressed IGF-1 in bone (*Figure 1—figure supplement 1a*) and circulating IGF-1 was statistically unchanged in *Bmp2* Prx1-cKO mice (*Figure 1—figure supplement 1b*). During ex vivo organ culture, WT and *Bmp2* Prx1-cKO metatarsals were equal in length on day one and both grew in culture (*Figure 1n*). WT and *Bmp2* Prx1-cKO metatarsals were equal in width on day 1, but only WT metatarsals grew in width during culture (*Figure 1o*).

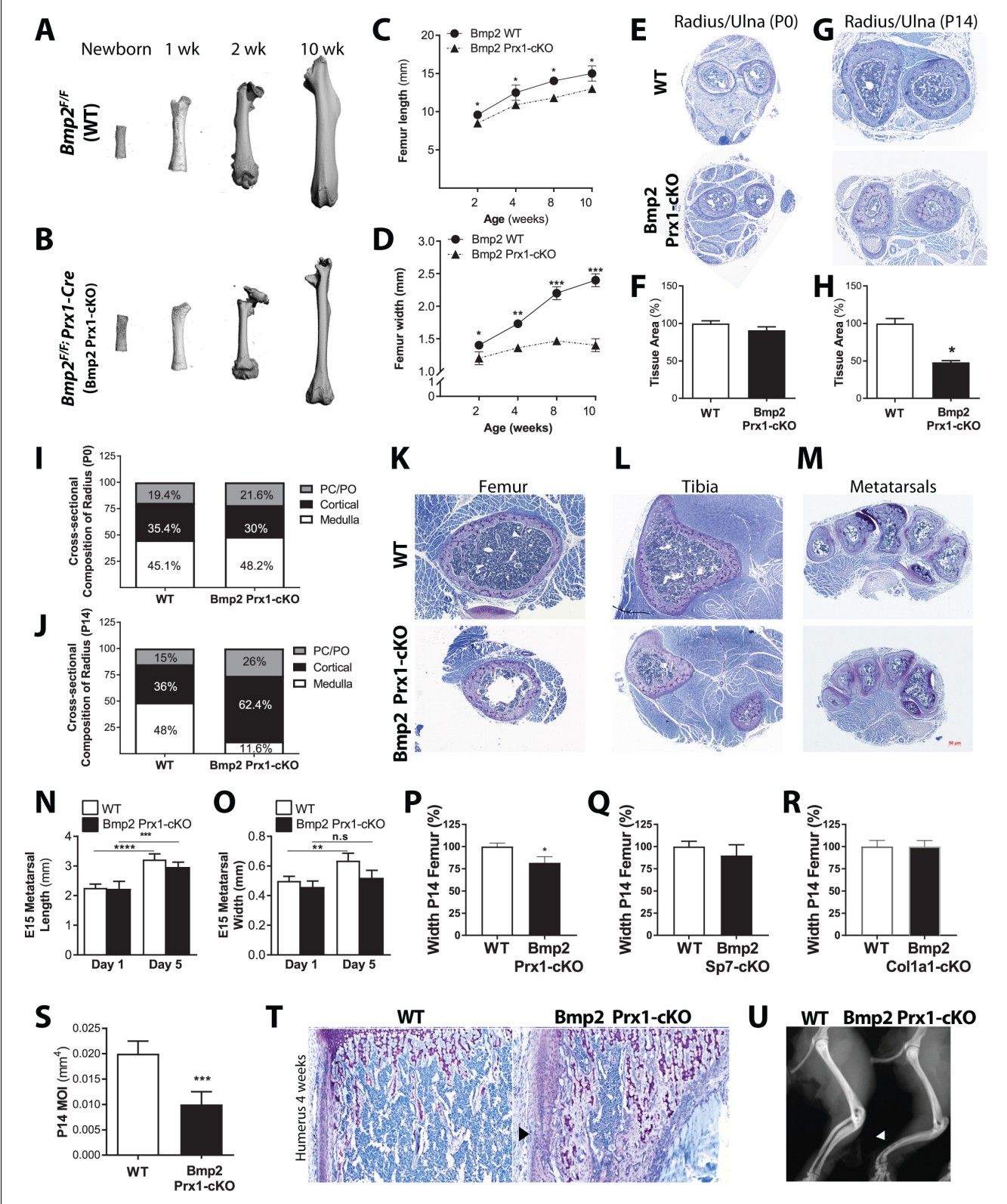

**Figure 1.** Osteoprogenitor-derived *Bmp2* couples length to width in the appendicular skeleton. (**a,b**) Representative 3D reconstructions of the murine femur using microcomputed tomography (microCT). (**c**) Femoral length or (**d**) femoral width at mid-diaphysis, presented as mean ± s.d. with *n* = 8–20 bones per age per genotype. *p<0.05, **p<0.005, or ***p<0.0005 vs. age-matched *Bmp2* Prx1-cKO cohort. (**e,g**) Representative toluidine blue histology at the mid-diaphysis of the forelimb. (**f,h**) MicroCT analysis of total cross-sectional bone tissue area presented as mean ±s.d. with *n* = 4.

*Figure 1 continued on next page*

*Figure 1 continued*

*p>0.05. (**i,j**) Cross-sectional composition of cortical bone in the radius of newborn (n = 6–9 per genotype) or 2 week-old mice (n = 3–9 per genotype) (see Materials and methods). Abbreviations: PC, perichondrium; PO, periosteum. (**k,l,m**) Representative toluidine blue histology at the mid-diaphysis of indicated skeletal elements. (**n**) Length or (**o**) width of embryonic day 15 metatarsals, measured following 1 or 5 days of ex vivo culture. Mean ±s.d. with n = 6–12 where **p<0.005, ***p<0.005, or ****p<0.00005. (**p,q,r**) Femur width mean ±s.d. with n = 6–12 where *p<0.05. (**s**) Minimum moment of inertia in P14 femur, calculated by microCT (n = 4) shown as mean ±s.d. where ***p<0.0005. (**t**) Toludine blue histology revealing cortical microcracks in the humerus of *Bmp2* Prx1-cKO mice at 4 weeks of age. (**u**) X-ray images showing representative bowing of the radius and ulna of *Bmp2* Prx1-cKO mice in the absence of frank fractures. Statistical analyses were performed using two-tailed Student's *t*-test.

DOI: https://doi.org/10.7554/eLife.42386.002

The following figure supplements are available for figure 1:

**Figure supplement 1.** BMP2 acts downstream of IGF-1 pathway in the periosteum.

DOI: https://doi.org/10.7554/eLife.42386.003

**Figure supplement 2.** Skeletal phenotype analysis of *Bmp2^Flox/Flox^*; *Col1a1-Cre* mice shows that loss of *Bmp2* in mature osteoblasts does not cause a periosteal growth defect.

DOI: https://doi.org/10.7554/eLife.42386.004

**Figure supplement 3.** Skeletal phenotype analysis of *Bmp2^Flox/Flox^*; *Prx1-Cre* mice reveals architectural abnormalities compounded by material defects.

DOI: https://doi.org/10.7554/eLife.42386.005

*Bmp2* is essential for conversion of *Runx2+* osteoprogenitors to the *Sp7+* osteoblast cell fate in primary cell models (*Salazar et al., 2016b*). Consistent with these in vitro observations, mice with conditional ablation of *Bmp2* in skeletal progenitors (*Bmp2* Prx1-cKO) (*Figure 1p*), but not committed osteoblasts (*Bmp2^Flox/Flox^*; *Sp7-GFP::Cre* or *Bmp2* Sp7-cKO) (*Figure 1q*) or mature osteoblasts (*Bmp2^Flox/Flox^*; *2.3kbCol1a1*-Cre or *Bmp2* Col1a1-cKO) (*Figure 1r*) showed periosteal growth defects at 2 weeks of age. *Bmp2* Col1a1-cKO mice exhibited no differences in skeletal phenotype when qualitatively examined by whole mount skeletal staining at birth (*Figure 1—figure supplement 2a–b*), histology of forelimb (*Figure 1—figure supplement 2c*) or hindlimb (*Figure 1—figure supplement 2d*) at 2 weeks of age, or X-ray imaging at 3 months of age (*Figure 1—figure supplement 2e*). Quantitative analysis of the femur at 2 weeks of age showed no changes in length, width (*Figure 1—figure supplement 2f*) or other standard parameters of trabecular bone volume fraction of the distal metaphysis and cortical cross-sectional area of the mid-shaft (*Table 1*).

**Table 1.** Skeletal phenotype analysis of *Bmp2^Flox/Flox^*; *Col1a1-Cre* mice shows that loss of *Bmp2* in mature osteoblasts does not cause a periosteal growth defect.

Bone mass analyzed in the femur of juvenile 2 week-old mice by microCT. Data presented as mean ±s.d. with no statistical differences detected between WT and conditional knockout mice using 1-way ANOVA. Abbreviations: BV/TV, trabecular bone volume to total tissue volume; Tb.Th, trabecular thickness; Tb.Sp. trabecular spacing; Tb.N. trabecular number; Tt.Ar, total cross-sectional tissue area at the mid-diaphysis; Ct.Ar, cortical bone area; Ct.Ar/Tt.Ar, cortical bone area as a fraction of total tissue area; C.Th cortical thickness; $I_{MIN}$, minimum moment of inertia.

| MicroCT femur, P14 | *Bmp2^Flox/Flox^* | *Bmp2^Flox/Flox^*; *Col1a1-Cre* |
|---|---|---|
| N | 4 | 4 |
| BV/TV (%) | 7.9 ± 1.0 | 7.5 ± 0.6 |
| Tb.Th (mm) | 0.026 ± 0.002 | 0.026 ± 0.006 |
| Tb.Sp. (mm) | 0.361 ± 0.037 | 0.399 ± 0.058 |
| Tb.N (1/mm) | 2.89 ± 0.307 | 2.63 ± 0.35 |
| Tt.Ar (mm$^2$) | 1.17 ± 0.08 | 1.15 ± 0.078 |
| Ct.Ar (mm$^2$) | 0.322 ± 0.052 | 0.301 ± 0.018 |
| Ct.Ar/Tt.Ar (%) | 27 ± 0.08 | 25 ± 0.08 |
| C.Th (mm) | 0.081 ± 0.006 | 0.078 ± 0.007 |
| $I$min (mm$^4$) | 0.040 ± 0.007 | 0.038 ± 0.002 |

DOI: https://doi.org/10.7554/eLife.42386.006

## Geometric abnormalities plus material defects underlie spontaneous fractures in *Bmp2*<sup>Flox/Flox</sup>; *Prx1-Cre* mice

Beyond the slender bone phenotype, *Bmp2* Prx1-cKO mice exhibited no differences in skeletal patterning when imaged by X-ray at 3 months of age (*Figure 1—figure supplement 3a*). Closer examination of *Bmp2* Prx1-cKO bones at 2 weeks of age by microCT revealed trabecular bone content and tissue mineral density were unaffected by the absence of *Bmp2* (*Figure 1—figure supplement 3b* and *Table 2*), despite significant reductions in bone width (*Figure 1—figure supplement 3c*), total cross-sectional area (Tt.Ar), marrow area (Ma.Ar), and minimum moment of inertia ($I_{MIN}$) (*Table 2*). Static histomorphometry disclosed decreased numbers of osteoblasts but not osteoclasts per unit of bone surface (*Figure 1—figure supplement 3d*). Bone formation and mineral apposition rates were similar at endosteal sites but dramatically reduced at periosteal sites in *Bmp2* Prx1-cKO mice (*Figure 1—figure supplement 3e–h*). We found abundant cortical porosity (*Figure 1—figure supplement 3i*) with residual islands of cartilage and excessive numbers of osteocytes in *Bmp2* Prx1-cKO bones (*Figure 1—figure supplement 3j*). Under polarizing light microcopy, collagen in *Bmp2* Prx1-cKO cortical bone had a woven appearance compared to lamellar organization in WT bone (*Figure 1—figure supplement 3k,l*). *Bmp2* Prx1-cKO bones had increased osteoid thickness and a prolonged mineralization lag time on periosteal but not endosteal surfaces (*Figure 1—figure supplement 3m–p*).

Overall, dramatically reduced polar moment of inertia (*Figure 1—figure supplement 1s*) and impaired material properties of *Bmp2* Prx1-cKO bones likely caused microcracks (*Figure 1t*) and bowing (*Figure 1u*), factors preceding onset of frank fractures (*Tsuji et al., 2006*).

## *Bmp2* is expressed at the right time and place to regulate periosteum formation and/or function

We used a bacterial *beta-galactosidase (lacz)* reporter expressed from the endogenous *Bmp2* locus (*Bmp2*<sup>lacz/+</sup>) to map the skeletal *Bmp2* expression domain (*Figure 2—figure supplement 1a–c*). At E13.5, LacZ was highly expressed in the ribs, scapula, clavicles, forelimbs, hindlimbs, and portions of the craniofacial vault (*Figure 2a*). At E14.5, LacZ+ cells surrounded the cartilage anlagen where the

**Table 2.** Skeletal phenotype of 2 week-old *Bmp2*<sup>Flox/Flox</sup>; *Prx1-Cre* mice.

Quantitative microCT data presented as mean ±s.d. where ***p<0.0005 vs. age-matched *Bmp2*<sup>F/F</sup> littermates when compared by 1-way ANOVA. (e) Static histomorphometry ($n$ = 4), presented as mean ±s.d. *p<0.05 vs. age-matched *Bmp2*<sup>F/F</sup> littermates. Abbreviations: BV/TV, trabecular bone volume to total tissue volume; Tb.Th, trabecular thickness; Tb.Sp. trabecular spacing; Tb.N. trabecular number; Tt.Ar, total cross-sectional tissue area at the mid-diaphysis; Ct.Ar, cortical bone area; Ct.Ar/Tt.Ar, cortical bone area as a fraction of total tissue area; C.Th cortical thickness; Ma.Ar, marrow area; $I_{MIN}$, minimum moment of inertia; TMD, tissue mineral density.

| MicroCT femur, P14 | *Bmp2*<sup>Flox/Flox</sup> | *Bmp2*<sup>Flox/Flox</sup>; *Prx1-Cre* |
| --- | --- | --- |
| N | 4 | 4 |
| BV/TV (%) | 4.3 ± 1.0 | 6.6 ± 0.9 |
| Tb.Th (mm) | 0.02 ± 0.001 | 0.02 ± 0.001 |
| Tb.Sp. (mm) | 0.35 ± 0.04 | 0.26 ± 0.06 |
| Tb.N (1/mm) | 2.8 ± 0.3 | 3.9 ± 1.0 |
| Tt.Ar (mm²) | 1.02 ± 0.08 | 0.6 ± 0.03*** |
| Ct.Ar (mm²) | 0.24 ± 0.03 | 0.21 ± 0.01 |
| Ct.Ar/Tt.Ar (%) | 23 ± 0.2 | 36 ± 1.0*** |
| C.Th (mm) | 0.065 ± 0.07 | 0.06 ± 0.03 |
| Ma.Ar (mm²) | 0.46 ± 0.04 | 0.23 ± 0.02*** |
| *I*min (mm⁴) | 0.02 ± 0.005 | 0.01 ± 0.001*** |
| TMD (mgHA/cm³) | 882.6 ± 20.6 | 858.7 ± 8.57 |

DOI: https://doi.org/10.7554/eLife.42386.007

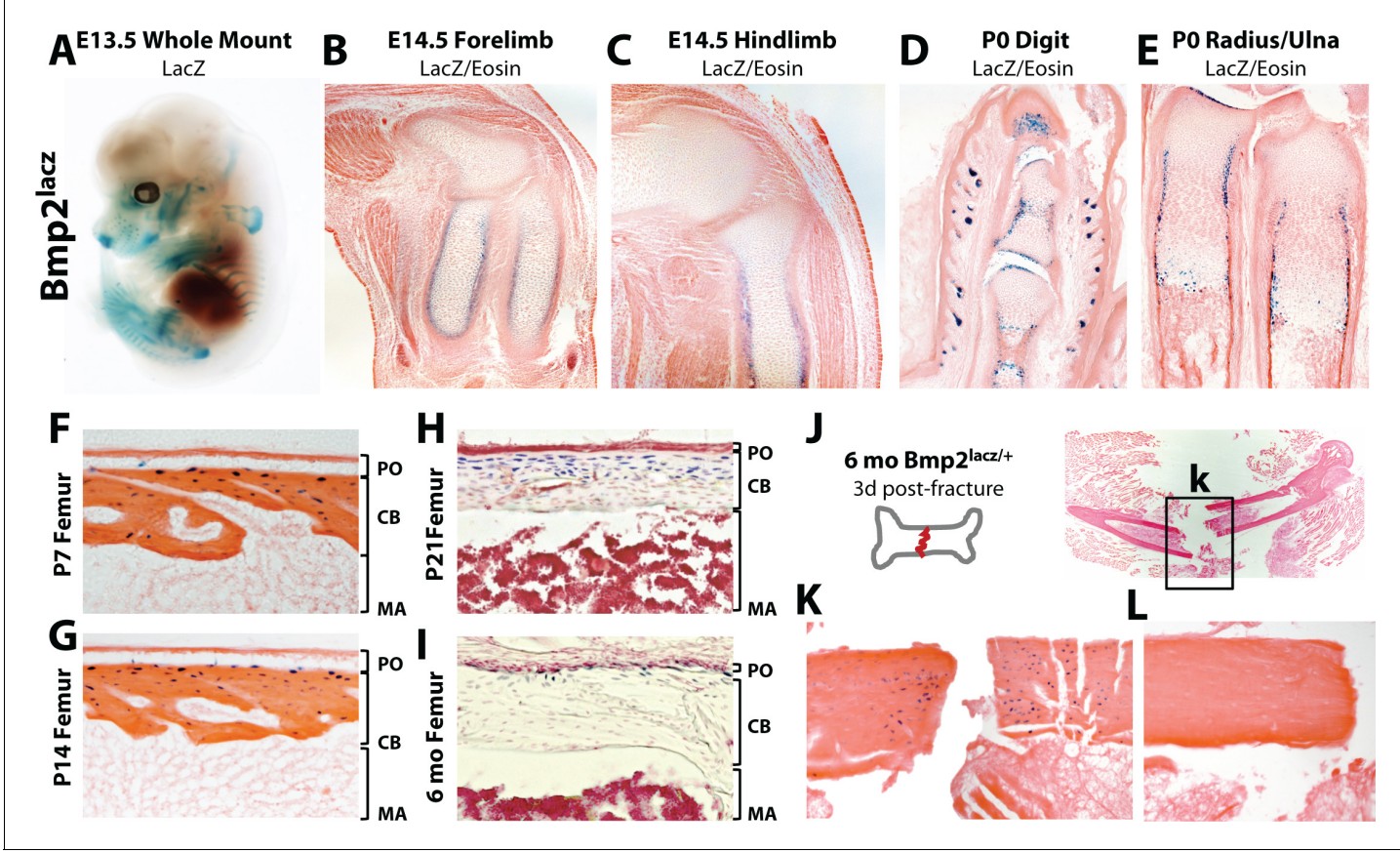

**Figure 2.** Robust versus quiescent states of *Bmp2* expression reflect active versus homeostatic states of periosteal bone growth. LacZ staining on tissues from mice expressing *beta-galactosidase* from one allele of the endogenous *Bmp2* locus (*Bmp2^lacz/+^*). (a) Lateral view of a whole mount E13.5 *Bmp2^lacz/+^* mouse embryo, representative of other *Bmp2^lacz/+^* littermates. (b,c) Longitudinal sections through the (b) forelimb or (c) hindlimb of E14.5 *Bmp2^lacz/+^* mouse embryos. (d,e) Longitudinal sections of (d) digits or (e) radius/ulna of newborn *Bmp2^lacz/+^* mice. (f–l) Longitudinal sections through cortical femoral bone from (f) 7 day-old, (g) 14 day-old, (h) 21 day-old or (i) 6 month-old *Bmp2^lacz/+^* mice. Abbreviations: PO (periosteum), CB (cortical bone), or MA (marrow) in brackets. (j–l) Standardized fractures were established in femurs of *Bmp2^lacz/+^* and WT littermate mice (*n* = 3). (k,l) LacZ staining 3 days post-fracture in (k) *Bmp2^lacz/+^* or (l) negative control WT mice.
DOI: https://doi.org/10.7554/eLife.42386.008

The following figure supplements are available for figure 2:

**Figure supplement 1.** Schematic of the *Bmp2^lacz^* knock-in allele.
DOI: https://doi.org/10.7554/eLife.42386.009

**Figure supplement 2.** *Bmp2* expression domain during periosteal bone growth.
DOI: https://doi.org/10.7554/eLife.42386.010

presumptive bone collar forms (*Figure 2b–c*, *Figure 2—figure supplement 2a*). At birth, LacZ+ cells populated the bone collar, perichondrium, hypertrophic cartilage and Groove of Ranvier (*Figure 2d–e*, *Figure 2—figure supplement 2b*). Although expecting to see strong LacZ activity in the periosteum during the first 2 weeks of life, we instead made the surprising observation that LacZ + cells in the periosteum were rare and appeared only in the cambium, immediately adjacent to or just below the outer bone surface (*Figure 2f–g*, *Figure 2—figure supplement 2c*). Most cortical osteocytes were LacZ+ while cells on the cortical endosteum were LacZ- (*Figure 2f–i*). LacZ activity became progressively restricted to the cortical/periosteal interface as mice approached peak body size and entered skeletal homeostasis (*Figure 2h–i*) but was reactivated locally following fracture (*Figure 2j–l*).

## Robust versus quiescent states of periosteal BMP signaling reflect active versus homeostatic states of periosteal bone growth

To compare active BMP signaling with sites of *Bmp2* expression, we utilized transgenic mice expressing enhanced green fluorescent protein (*gfp*) controlled by a pan-BMP-responsive promoter element of the *Id1* gene (*BRE:gfp*) (*Monteiro et al., 2008*). At E13.5, GFP+ cells flanked cartilage rudiments of the forelimb (*Figure 3a*). In newborn forelimbs, GFP+ cells co-localized with mineralized bone (*Figure 3b*), the entire length of the newly-formed bone collar (*Figure 3c*, white arrows and red dashed line), as well as hypertrophic cells in growth plate cartilage (*Figure 3c*, open red arrows). In hindlimbs at 2 weeks, GFP+ cells surrounded trabecular structures in the medullary cavity and populated the inner layer of the periosteum (*Figure 3d,e*, red dashed line). By six months, when

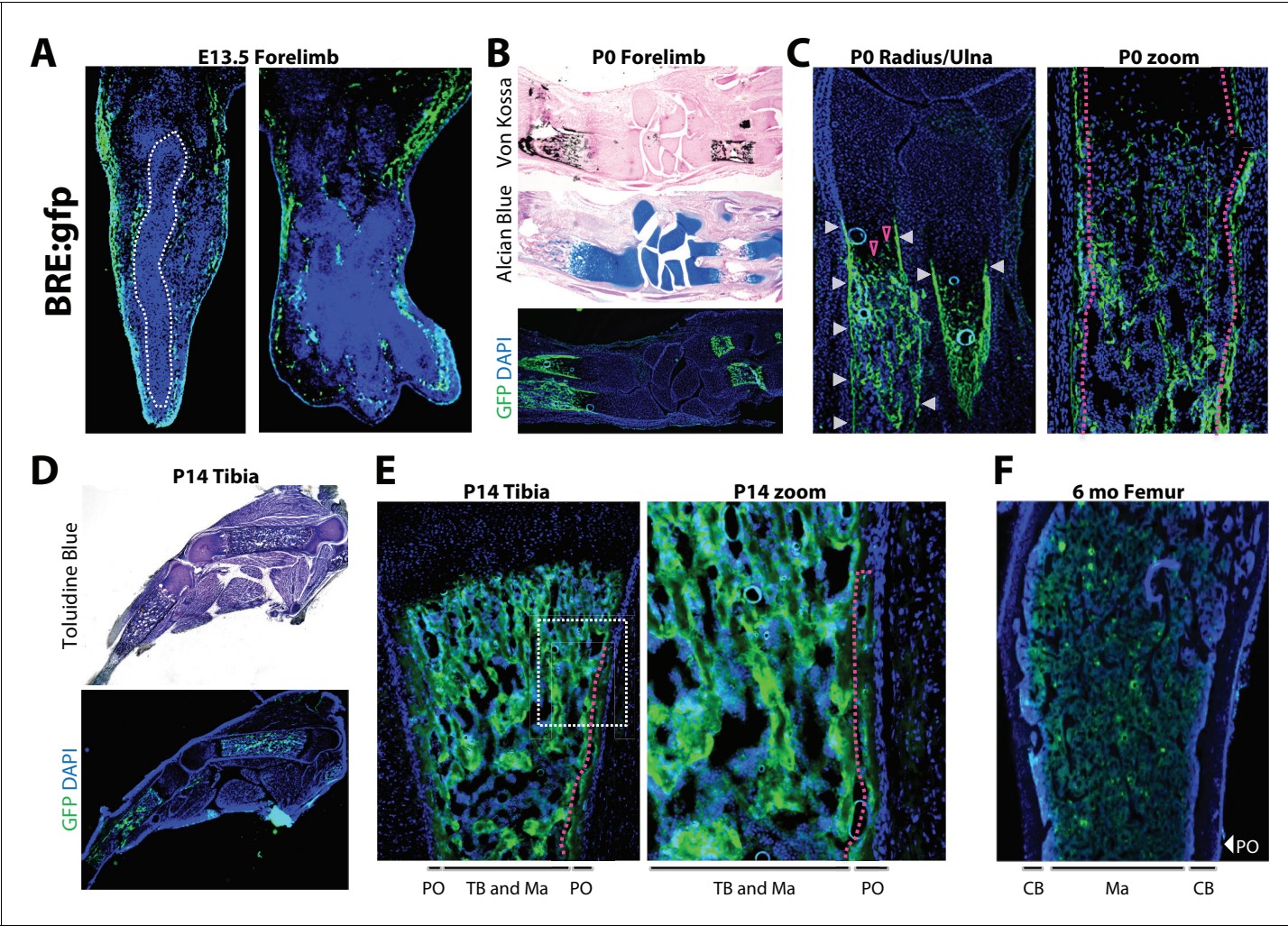

**Figure 3.** Robust versus quiescent states of periosteal BMP signaling reflect active versus homeostatic states of periosteal bone growth. Fluorescent and brightfield microscopy on tissues from mice with transgenic expression of *enhanced green fluorescent protein (gfp)* under the control of a minimal fragment of the *Id1* promoter with pan-BMP response elements (*BRE:gfp*). (a) GFP (green) and DAPI (blue) imaging on sagittal (left) or frontal (right) cryosections of the hand plate from E13.5 *BRE:gfp* embryos. (b,c) Serial sections through the forelimb of newborn *BRE:gfp* mice were analyzed by Von Kossa staining for mineralized tissue, Alcian Blue staining for cartilage, or DAPI counterstaining of GFP expression domains. (d,e) Toluidine blue or GFP/DAPI imaging on sagittal cryosections through the femur of (d,e) 2 week-old or (f) 6 month-old *BRE:gfp* mice. White arrowheads, GFP in the bone collar surrounding the growth plate cartilage; red arrowheads, GFP+ cells in growth plate; red dotted lines demarcate GFP+ cells of the (2c) bone collar or (2e) innermost layer of the periosteum. Abbreviations: PO, periosteum; CB, cortical bone; TB, trabecular bone; Ma, Marrow. For all timepoints, $n \geq 3$ histological sections were examined from equivalent skeletal sites of multiple littermate mice.
DOI: https://doi.org/10.7554/eLife.42386.011

mice attained peak skeletal size, GFP expression had become quiescent at bone surfaces and was restricted to pockets of cells in the marrow (*Figure 3f*).

Cells with active BMP pathway are therefore abundant in newly-forming bone at mid-gestation and neonatal stages. A distinct GFP+ population demarcating the bone collar at birth transitions to a robust GFP+ inner periosteal layer flanking the diaphysis by 2 weeks of age, and subsequently remains as a sparsely GFP+ periosteal population in adult bones at homeostasis.

Importantly, *BRE:gfp* is not specific for BMP2 signaling, but rather integrates the net response of a cell to all local BMPs and BMP antagonists. Consistent with the cortical but not trabecular skeletal phenotype of *Bmp2* Prx1-cKO mice (*Figure 1a–u*), this comparative analysis of *Bmp2-lacz* versus *BRE:gfp* strongly suggests BMP2 is the major BMP family member driving BMP signaling at periosteal but not necessarily trabecular sites.

## *Bmp2* is required for BMP signaling and expression of BMP target genes in the periosteum

We next determined if the periosteum forms in *Bmp2* Prx1-cKO mice and if it engages in BMP signaling. Picrosirius red histology revealed that embryonic development and postnatal maintenance of the periosteum occur independently of *Bmp2* expression in the *Prx1-Cre* lineage (*Figure 4a–c*). By contrast, BMP2 is necessary for phospho-activation of Smads1/5 at inner periosteal but not endosteal bone surfaces. This was evident at birth (*Figure 5a*) and became pronounced by 2 weeks of age (*Figure 5b*). Immunohistochemistry for ID1 protein, a BMP-target gene, showed that ID1+ cells were abundant in the periosteum of WT but not *Bmp2* Prx1-cKO mice (*Figure 5c*). We performed in situ hybridization for *Col1a1*, encoding the major extracellular matrix protein produced by osteoblasts. Wild-type but not *Bmp2* Prx1-cKO mice expressed *Col1a1* in the periosteum (*Figure 5d*). Periosteal BMP2 signaling is therefore necessary for osteoblast specification in the periosteal niche.

## *Bmp2* acts downstream of intermittent PTH therapy in the periosteum

Intermittent parathyroid hormone (PTH) therapy and sclerostin neutralizing antibody (SOST-ab) are two successful clinical approaches to stimulate the periosteum for periosteal growth and accelerated fracture repair (*Baron and Kneissel, 2013*; *Collinge and Favela, 2016*; *Einhorn and Gerstenfeld, 2015*). PTH is a naturally occurring hormone that regulates mineral homeostasis via endocrine actions on multiple organs including bone. Intermittent exposure to PTH (iPTH) through once daily injections of $hPTH_{1-34}$ peptide results in a net surplus of new bone formation (*Neer et al., 2001*). The mechanism of action for iPTH therapy continues to generate considerable discussion and PTH has been reported to induce expression of *Bmp2* in osteoblast cultures (*Zhang et al., 2011*). To examine the interaction between PTH and *Bmp2* in the periosteum, we treated 2 week-old *Bmp2* Prx1-cKO mice with iPTH for 14 days and monitored periosteal bone growth in the now 1 month-old treated juveniles (*Figure 6a*). At 2 weeks of age, WT and *Bmp2* Prx1-cKO mice responded to iPTH with increased trabecular bone mass (*Figure 6b,d,f* and *Table 3*). WT but not *Bmp2* Prx1-cKO mice exhibited a trend for periosteal expansion (*Figure 6c,e,g* and *Table 3*), although this was not statistically significant and it is plausible that additional treatment time was necessary to achieve sufficient cumulative periosteal growth to become measurable by microCT. WT and *Bmp2* Prx1-cKO mice had elevated serum calcium and reduced serum phosphorus levels following iPTH (*Figure 6l,m*). Bone formation rate and mineral apposition rate were significantly blunted at periosteal but not endosteal surfaces of cortical bone in *Bmp2* Prx1-cKO mice, and were not elevated to the levels achieved in WT mice following treatment with iPTH (*Figure 6h–k*). WT but not *Bmp2* Prx1-cKO mice treated with iPTH expressed ID3, a BMP target gene, in the periosteum (*Figure 6n*), whereas both genotypes had ID3+ cells in endosteal compartments (*Figure 6n*, blue lines).

In adults treated for two weeks with iPTH (*Figure 7a*), WT and *Bmp2* Prx1-cKO mice had increased trabecular bone mass following iPTH (*Figure 7b* and *Table 4*). To evaluate biomechanical properties in the femur, peak moment (maximum load during failure in three-point bending (*Makowski et al., 2014*)) was plotted as a function of $I_{MIN}/C_{MIN}$. WT mice experienced much greater cortical expansion with iPTH than *Bmp2* Prx1-cKO mice, as evident by cross-sectional ratio proportional to bending moment (*Figure 7c*), and moment of inertia that is predominantly influenced by the periosteal perimeter (*Table 4*). Functionally, iPTH-treated *Bmp2* Prx1-cKO mice were 2–4 times more likely than non-iPTH treated knockouts to experience femoral fracture (*Figure 7e*). Forelimb

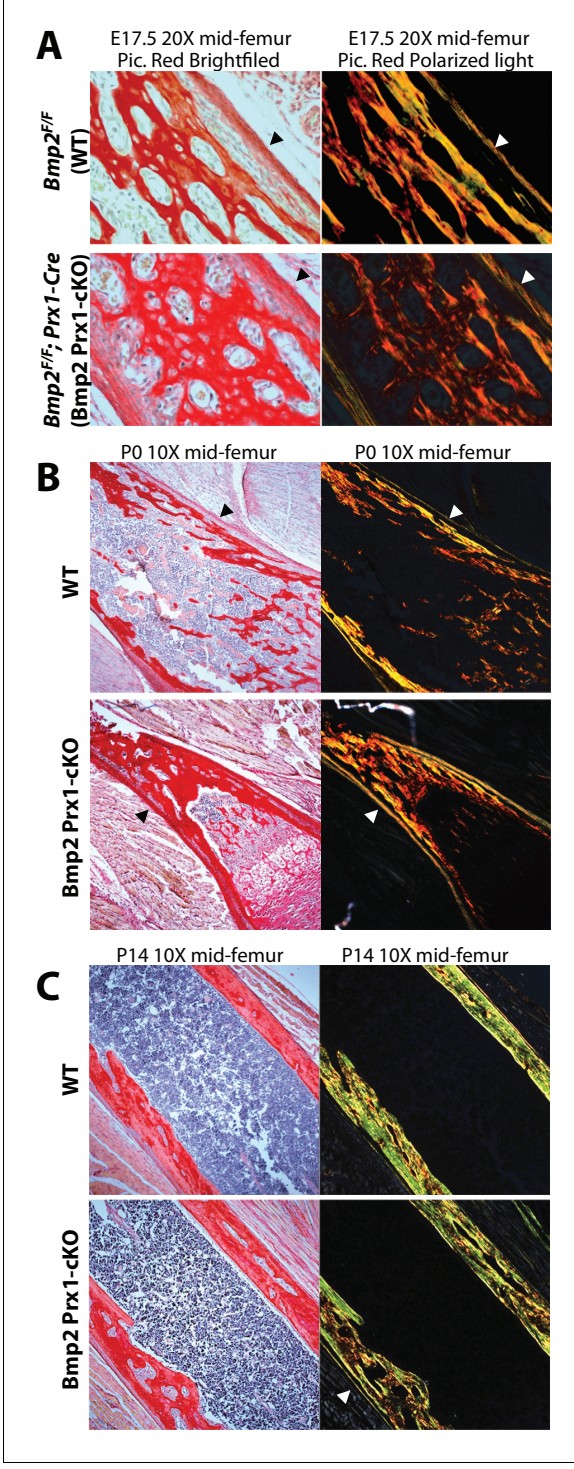

**Figure 4.** *Bmp2* is dispensable for development and maintenance of the periosteum. (**a,b**) The periosteum forms during development and is maintained in postnatal life in *Bmp2* Prx1-cKO mice. Sagittal sections of the femur from mice were stained with picrosirius red and hematoxylin, and imaged by brightfield (left panels) or polarized light (right panels) microscopy. Images shown are representative of femurs harvested from littermates at ages (**a**) embryonic day 17.5, (**b**) postnatal day 0, or (**c**) postnatal day 14. Arrowheads point to the outer collagen-rich canopy of periosteum in *Bmp2* Prx1-cKO mice.
DOI: https://doi.org/10.7554/eLife.42386.012

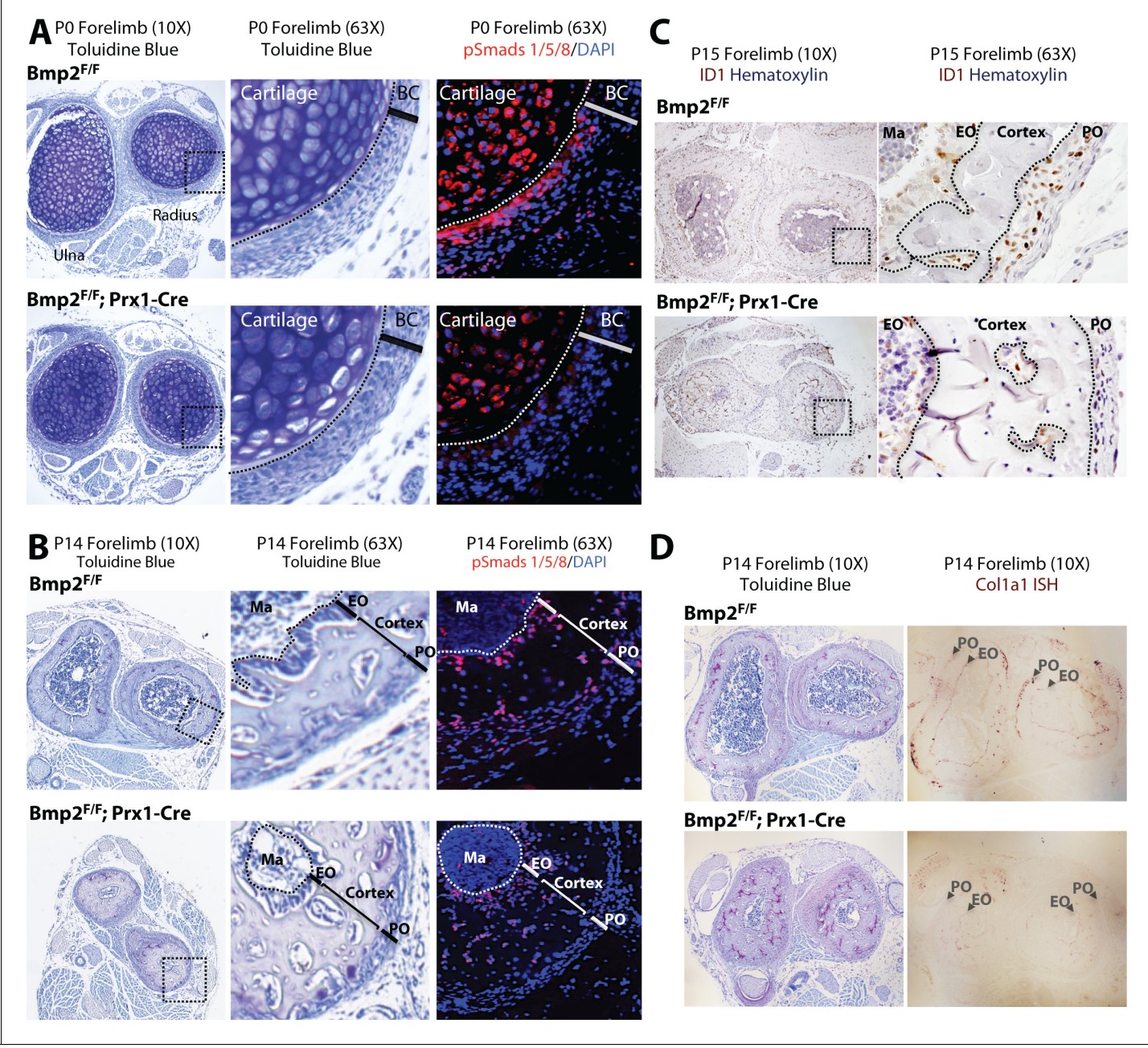

**Figure 5.** *Bmp2* is essential for periosteal BMP signaling and periosteal expression of BMP target genes. (a,b) Loss of phospho-Smad1/5+ cells in *Bmp2* Prx1-cKO periosteum. Transverse serial sections of the radius and ulna from (a) newborn and (b) 2 week-old mice, imaged in brightfield following toluidine blue stain to visualize skeletal tissue (left panels) or by fluorescence microscopy following DAPI and immunostaining to visualize cells with phospho-activated Smads1/5 (right panels). Black boxes on left panels indicate regions expanded in two right panels. (c) Loss of ID1+ cells in *Bmp2* Prx1-cKO periosteum. Transverse sections of the radius and ulna were imaged in brightfield following immunostaining to visualize cells expressing the BMP target gene, *Id1*. Black boxes on left panels indicate regions expanded in right panels. (d) Toluidine blue and in situ hybridization for *Col1a1* in cross-sections of the radius/ulna from 2 week-old *Bmp2* Prx1-cKO mice. Abbreviations: BC, bone collar/perichondrium; EO, endosteum; PO, periosteum; Ma, marrow. For all timepoints, $n \geq 3$ histological sections were examined from equivalent skeletal sites of multiple littermate mice.
DOI: https://doi.org/10.7554/eLife.42386.013

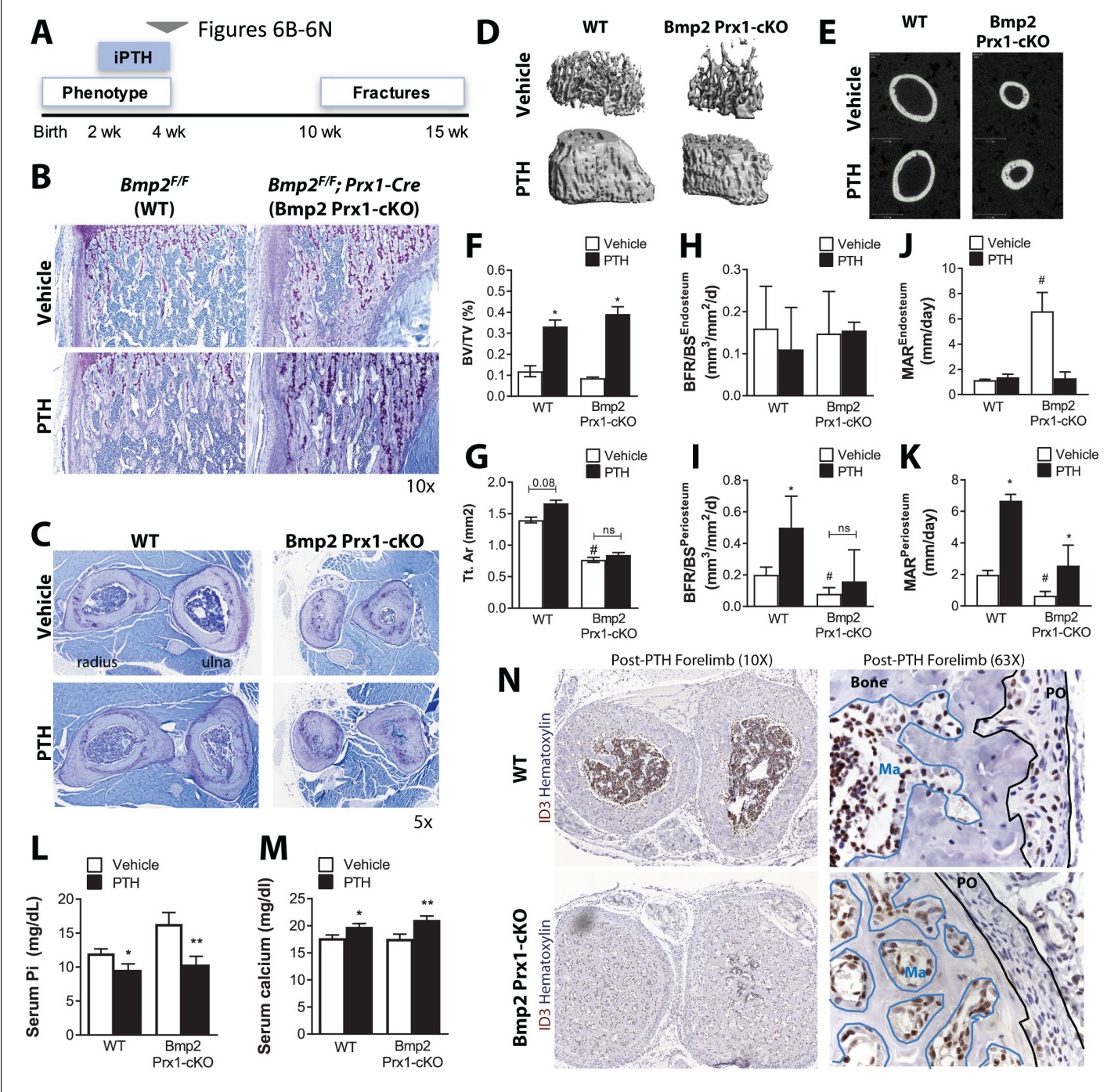

**Figure 6.** *Bmp2* acts downstream of intermittent parathyroid hormone treatment in the juvenile periosteum. Intermittent PTH$_{1-34}$ therapy does not rescue periosteal growth in juvenile *Bmp2* Prx1-cKO mice. (**a**) Juvenile mice were given intermittent PTH$_{1-34}$ therapy (100 mg/kg, subcutaneous) for 14 days. (**b**) Longitudinal sections of the femur stained with toluidine blue to visualize trabecular bone architecture. (**c**) Transverse sections of the radius and ulna stained with toluidine blue to visualize cortical bone architecture. (**d–g**) Bone mass analyzed in the femur by microcomputed tomography (microCT). (**d**) Trabecular bone at the distal metaphysis and (**e**) cortical bone at the mid-diaphysis of the femur visualized by 3D reconstructions. Images represent the group mean and are shown to scale. (**f**) Ratio of bone volume (BV) to trabecular volume (TV). (**g**) Total cross-sectional area at the mid-diaphysis. Quantitative microCT data presented as mean ±s.d. where *p<0.05 vs. matched genotype vehicle control and #p<0.05 vs. WT vehicle control (*n* = 4–5 per group). (**h,i**) Dynamic histomorphometry assessing bone formation rate as a function of bone surface (BFR/BS) at (**h**) endosteal versus (**i**) periosteal surfaces. (**j,k**) Dynamic histomorphometry assessing mineral apposition rate (MAR) at (**j**) endosteal versus (**k**) periosteal surfaces. Dynamic histomorphometry (*n* = 4–5 per group) presented as mean ±s.d. where *p<0.05. BFR$^{PO}$ *P*-value=0.0503) vs. matched genotype vehicle control and #p<0.05 vs. WT vehicle control. (**l,m**) Elisa analysis measuring circulating (**l**) serum phosphate and (**m**) serum calcium in juvenile mice treated with

*Figure 6 continued on next page*

*Figure 6 continued*

intermittent PTH$_{1-34}$ presented as mean ±s.d. where *p<0.05 vs. vehicle-treated *Bmp2*$^{F/F}$ or **p<0.05 vehicle-treated *Bmp2* Prx1-cKO littermates. (**n**) Transverse sections of the radius/ulna with immunostaining to visualize cells expressing the BMP target gene, ID3. Abbreviations: PO, periosteum; Ma, marrow. $n \geq 3$ histological sections were examined from multiple mice per cohort.

DOI: https://doi.org/10.7554/eLife.42386.014

fractures in *Bmp2* Prx1-cKO mice treated with iPTH shifted from unilateral to bilateral occurrence (*Figure 7f,g*). Fracture incidence was not improved by increasing iPTH therapy to 3 weeks. Existing fractures remained unhealed (*Figure 7h*).

Expression of *Bmp2* in the *Prx1-Cre* lineage is therefore a major if not essential contributor to the mechanism of action by which iPTH stimulates the periosteum for periosteal bone growth and conclusively essential for the mechanism by which PTH improves bending strength and stimulates the periosteum for fracture repair.

### *Bmp2* acts downstream of canonical WNT signaling in the periosteum

Haploinsufficiency of *Dkk1*, a secreted antagonist of LRP5/6, enhances canonical WNT signaling and induces high bone mass in mice (*Morvan et al., 2006*), findings that prompted efforts to develop DKK1 neutralizing antibodies for systemic activation of bone formation in clinical orthopedic settings (*Ke et al., 2012*). Haploinsufficiency of *Dkk1* (*Dkk1$^{+/-}$*) was recently reported to have no effect on the formation or healing of spontaneous fractures in adult mice lacking *Bmp2* (*Dkk1$^{+/-}$; Bmp2$^{Flox/Flox}$; Prx1-Cre*) (*Intini and Nyman, 2015*). However, DKK1 neutralizing antibodies have been shown to enhance bone formation in younger animals to a greater degree than in older animals (*Ke et al., 2012*). We therefore revisited this genetic mouse model to determine whether activation of WNT signaling through haploinsufficiency of *Dkk1* could restore periosteal bone growth in young *Bmp2* Prx1-cKO mice, when the skeletal phenotype is first established and thus prior to the onset of fractures (*Figure 8a*). By 2 weeks of age, trabecular bone mass was elevated by haploinsufficiency of

**Table 3.** Skeletal phenotype of juvenile *Bmp2$^{Flox/Flox}$; Prx1-Cre* mice after intermittent PTH therapy.

Juvenile mice (two weeks-old) were given intermittent PTH$_{1-34}$ therapy (100 mg/kg, subcutaneous) for 14 days. Bone mass was analyzed in the femur by microcomputed tomography (microCT). Trabecular bone at the distal metaphysis and cortical bone at the mid-diaphysis of the femur are presented as group mean ± s.d. and statistically compared by 2-way ANOVA. BV/TV, bone volume fraction; Conn. D, connectivity density; SMI, structure model index; Tb.N, trabecular number; Tb.Th, trabecular thickness; Tb.Sp, Trabecular separation; Tt.Ar, total cross-sectional area; Ct.Ar, cortical bone area; Ct.Ar/Tt.Ar, cortical area fraction; Ct.Th, average cortical thickness; *I*min, minimum moment of inertia; Ma.V, marrow volume. $P^a \leq 0.05$ vs. WT. $P^b \leq 0.05$ vs. Vehicle.

| MicroCT femur, four wk | *Bmp2$^{Flox/Flox}$* | | *Bmp2$^{Flox/Flox}$; Prx1-Cre* | |
|---|---|---|---|---|
| Treatment | Vehicle | PTH | Vehicle | PTH |
| N | 4 | 5 | 5 | 5 |
| BV/TV (%) | 11.9 ± 7.4 | 33.2 ± 8.6[b] | 8.7 ± 1.2 | 39.2 ± 9.6[b] |
| Conn.D | 244.6 ± 232.1 | 438.7 ± 165 | 132.2 ± 60.5 | 702.8 ± 168[b] |
| SMI | 2.2 ± 0.9 | 0.1 ± 1.8 | 2.7 ± 0.2 | −0.6 ± 1.0[b] |
| Tb.N (1/mm) | 5.1 ± 2.1 | 8.3 ± 2.4[b] | 4.8 ± 0.5 | 10.8 ± 1.3[b] |
| Tb.Th (mm) | 0.04 ± 0.005 | 0.05 ± 0.009 | 0.034 ± 0.002[a] | 0.051 ± 0.006[b] |
| Tb.Sp (mm) | 0.2 ± 0.07 | 0.121 ± 0.05 | 0.2 ± 0.02 | 0.08 ± 0.02[b] |
| Ct.Ar (mm$^2$) | 0.43 ± 0.04 | 0.60 ± 0.05[b] | 0.4 ± 0.03 | 0.5 ± 0.05[b] |
| Tt.Ar (mm$^2$) | 1.4 ± 0.1 | 1.6 ± 0.2 | 0.7 ± 0.09[a] | 0.8 ± 0.1 |
| Ct.Ar/Tt.Ar (%) | 0.31 ± 0.01 | 0.38 ± 0.02[b] | 0.57 ± 0.02[a] | 0.63 ± 0.05 |
| Ct.Th (mm) | 0.1 ± 0.007 | 0.134 ± 0.004[b] | 0.15 ± 0.015[a] | 0.19 ± 0.01[b] |
| Ma.V (mm$^3$) | 1.16 ± 0.7 | 1.2 ± 0.2 | 0.42 ± 0.09[a] | 0.37 ± 0.07 |
| *I*min (mm$^4$) | 0.06 ± 0.009 | 0.1 ± 0.02[b] | 0.03 ± 0.006[a] | 0.04 ± 0.01 |

DOI: https://doi.org/10.7554/eLife.42386.015

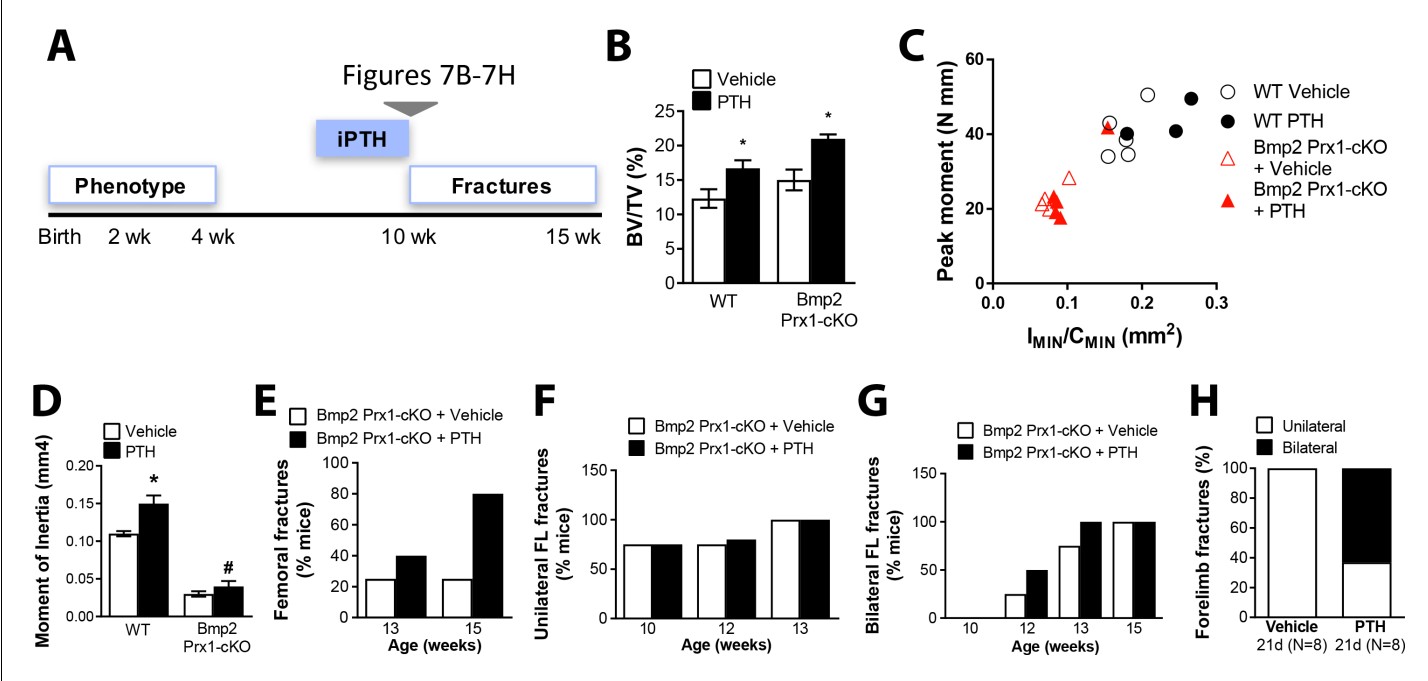

**Figure 7.** *Bmp2* acts downstream of intermittent parathyroid hormone treatment in the adult periosteum. Intermittent PTH$_{1-34}$ therapy does not improve biomechanical stability or fracture repair in adult *Bmp2* Prx1-cKO mice. (a) Adult mice were given intermittent PTH$_{1-34}$ therapy (100 mg/kg, subcutaneous) for (**b–g**) 14 days or (**h**) 21 days. (**b–d**) Quantitative microCT and biomechanical analysis on adult mice treated 2 weeks with PTH$_{1-34}$ (*n* = 3–5), presented as mean ±s.d. where *p<0.05 vs. matched genotype vehicle control. (b) Ratio of bone volume (BV) to trabecular volume (TV). (c) Peak moment as a function of I$_{MIN}$/C$_{MIN}$. (d) Predicted minimum moment of inertia. (**e–h**) Incidence of femoral or forelimb fractures in adult mice treated (**e–g**) two weeks (*n* = 4–5 per group) or (**h**) 21 days (*n* = 8 per group) with PTH$_{1-34}$. Biomechanics are reported as mean ±s.d, where *P*-value was calculated using 1-way ANOVA and post-test Newman Keulus. All data points were included in the analysis. Remaining group comparisons were made by 2-way ANOVA and Tukey's multiple comparison tests.

DOI: https://doi.org/10.7554/eLife.42386.016

*Dkk1*, and this was not dependent on *Bmp2* (data not shown). *Bmp2* Prx1-cKO mice developed a ˜75% decrease in calculated cross-sectional moment of inertia at the femoral mid-diaphysis that was neither rescued nor altered by haploinsufficiency of *Dkk1* (*Figure 8b–c*).

Since haploinsufficiency of *Dkk1* did not affect periosteal growth in young (*Figure 8a–c*) or aged mice (*Intini and Nyman, 2015*), regardless of *Bmp2* status, we chose sclerostin neutralizing antibody (SOST-ab) as an alternative method to stimulate canonical WNT signaling in bone and thereby activate the periosteum. Adult mice were treated for 2 weeks with SOST-ab (*Figure 8d*). WT males treated with SOST-ab had a significant increase in trabecular bone (almost 88%). *Bmp2* Prx1-cKO male mice undergoing SOST-ab treatment had ˜32% more trabecular bone than saline treated knockouts, an anabolic trend that reached statistical significance in females but not males (*Figure 8e–f*, and Source Data). At the mid-diaphysis (*Figure 8g–j*), WT mice treated with SOST-ab significantly enhanced total cross-sectional area (p=0.01; *Figure 8h*), minimum moment of inertia (p<0.0001; *Figure 8i*), and polar moment of inertia (p=0.0004; *Figure 8j*) when compared to non-treated genotype-matched and sex-matched controls. These parameters of cortical bone mass and strength were unchanged in *Bmp2* Prx1-cKO mice treated with SOST-ab (*Figure 8g–j* and Source Data). The number of forelimb fractures in *Bmp2* Prx1-cKO mice was not reduced by SOST-ab (*Figure 8k*) and these fractures showed no radiographic evidence of mineralized bridging (*Figure 8l*).

The abundance of phospho-activated Smads1/5 in bone lysates (*Figure 9a*) and alkaline phosphatase activity in the periosteum (*Figure 9b*) were increased 24 hr after one injection of PTH or SOST-ab. Crossing the *BRE:gfp* reporter into a *Bmp2* Prx1-cKO background revealed that SOST-ab dramatically upregulates periosteal BMP signaling in WT but not *Bmp2* Prx1-cKO mice, an effect evident after 72 hr (*Figure 9c* and *Figure 9—figure supplement 1*) or two weeks of treatment

**Table 4.** Skeletal phenotype of adult *Bmp2*<sup></sup>*Flox/Flox*; *Prx1-Cre* mice after intermittent PTH therapy.

Ten week-old mice were given intermittent PTH$_{1-34}$ therapy (100 mg/kg, subcutaneous) for 14 days. Bone mass was analyzed in the femur by microcomputed tomography (microCT). Trabecular bone at the distal metaphysis and cortical bone at the mid-diaphysis of the femur are presented as group mean ±s.d. and statistically compared by 2-way ANOVA. BV/TV, bone volume fraction; Tb.N, trabecular number; Tb.Th, trabecular thickness; Tb.Sp, Trabecular separation; Tt.Ar, total cross-sectional area; Ct.Ar, cortical bone area; Ct.Ar/Tt.Ar, cortical area fraction; Ct.Th, average cortical thickness; Ct.Po, cortical porosity; Ma.V, marrow volume; *I*min, minimum moment of inertia. $P^a \leq 0.05$ vs. WT. $P^b \leq 0.05$ vs. Vehicle.

| MicroCT Femur, 10 wk | *Bmp2*<sup>*Flox/Flox*</sup> | | *Bmp2*<sup>*Flox/Flox*</sup>; *Prx1-Cre* | |
|---|---|---|---|---|
| Treatment | Vehicle | PTH | Vehicle | PTH |
| N | 5 | 5 | 5 | 5 |
| BV/TV (%) | 12.3 ± 3 | 16.7 ± 2[b] | 15.0 ± 3.0 | 21.0 ± 1.4[b] |
| Tb.N (1/mm) | 4.90 ± 0.5 | 5.50 ± 0.9 | 6.10 ± 0.7[a] | 5.70 ± 0.6 |
| Tb.Th (mm) | 0.04 ± 0.003 | 0.04 ± 0.005 | 0.04 ± 0.001 | 0.05 ± 0.003[b] |
| Tb.Sp (mm) | 0.20 ± 0.02 | 0.19 ± 0.05 | 0.16 ± 0.02 | 0.16 ± 0.02 |
| Tt.Ar (mm$^2$) | 1.60 ± 0.01 | 1.90 ± 0.3 | 0.80 ± 0.08[a] | 0.90 ± 0.18 |
| Ct.Ar (mm$^2$) | 0.76 ± 0.07 | 0.88 ± 0.07[b] | 0.58 ± 0.06[a] | 0.72 ± 0.1b |
| Ct.Ar/Tt.Ar (%) | 0.47 ± 0.04 | 0.46 ± 0.03 | 0.75 ± 0.01 | 0.80 ± 0.027 |
| Ct.Th (mm) | 0.19 ± 0.02 | 0.19 ± 0.005 | 0.25 ± 0.01[a] | 0.30 ± 0.014[b] |
| Ct. Po (%) | 3.58 ± 0.28 | 4.06 ± 0.1[b] | 3.38 ± 0.3 | 2.80 ± 0.18[b] |
| Ma.V (mm$^3$) | 1.03 ± 0.16 | 1.21 ± 0.27 | 0.23 ± 0.03[a] | 0.21 ± 0.07[b] |
| *I*min (mm$^4$) | 0.11 ± 0.01 | 0.15 ± 0.03[b] | 0.03 ± 0.01[a] | 0.04 ± 0.02 |

DOI: https://doi.org/10.7554/eLife.42386.017

(*Figure 9d*). The periosteum of *Bmp2*<sup>*Flox/Flox*</sup>; *tdTomato* <sup>+/*Flox*</sup>; *Prx1-Cre* mice was populated by tdTomato+ cells despite this lack of periosteal response to SOST-ab (*Figure 9d* and *Figure 9—figure supplement 2*), consistent with our model in which the periosteum is present but not activated (*Figure 3a–f*) without complementation by BMP2 (*Chappuis et al., 2012*).

Expression of *Bmp2* in the *Prx1-Cre* lineage is therefore essential to the mechanism of action by which SOST-ab stimulates the periosteum for periosteal bone growth and fracture repair.

### *Bmp2* functions downstream of canonical WNT signaling in the osteoblast gene regulatory network

Periosteal cells isolated from long bones of 2 week-old mice express *Bmp2* and *Bmp7* mRNAs, but nearly undetectable levels of *Bmp4*. Cre-mediated deletion of *Bmp2* is highly efficient in *Bmp2* Prx1-cKO cells and does not induce *Bmp7* or *Bmp4* to compensate (*Figure 10a,b*). Primary periosteal cells from *Bmp2* Prx1-cKO mice were unable to be expanded or maintained without complementation by recombinant BMP2. Cells from the bone marrow stroma (BMSC) or embryonic mouse limb bud were therefore employed for mechanistic studies since we have previously demonstrated that unresponsiveness of *Bmp2*-deficient osteoprogenitors to canonical WNT signaling can be recapitulated using these in vitro models (*Salazar et al., 2016b*). In BMSC, an optimal ratio of BMP to WNT signaling must be maintained for osteoblast differentiation (*Figure 10c,d*). *Bmp2* Prx1-cKO cells exposed to Wnt3a do not upregulate *Sp7* (a transcription factor required for osteoblast specification (*Rodda and McMahon, 2006*; *Nakashima et al., 2002*) unless supplemented with recombinant BMP2 (*Salazar et al., 2016b*). Consistent with the observation that *Bmp7* is expressed (*Figure 10b*) but does not compensate for lack of *Bmp2* in periosteal cells in vivo, equivalent concentrations of recombinant BMP7 and GDF5 did not functionally compensate in vitro for lack of BMP2 in BMSC (*Figure 10e*).

*Bmp2* Prx1-cKO cells exposed to Wnt3a also do not upregulate Grainyhead-like 3 (*Grhl3*), a transcription factor co-expressed with *Bmp2* and *Lrp5/6* during embryonic skeletal development and fracture repair in adult bones (*Salazar et al., 2016b*). In vitro analysis suggested Grhl3 is necessary and sufficient for induction of *Sp7* by BMP2 (*Salazar et al., 2016b*). We thus investigated the relationship between WNT, BMP2, *Sp7*, and *Grhl3* in the osteoblast gene regulatory network and used

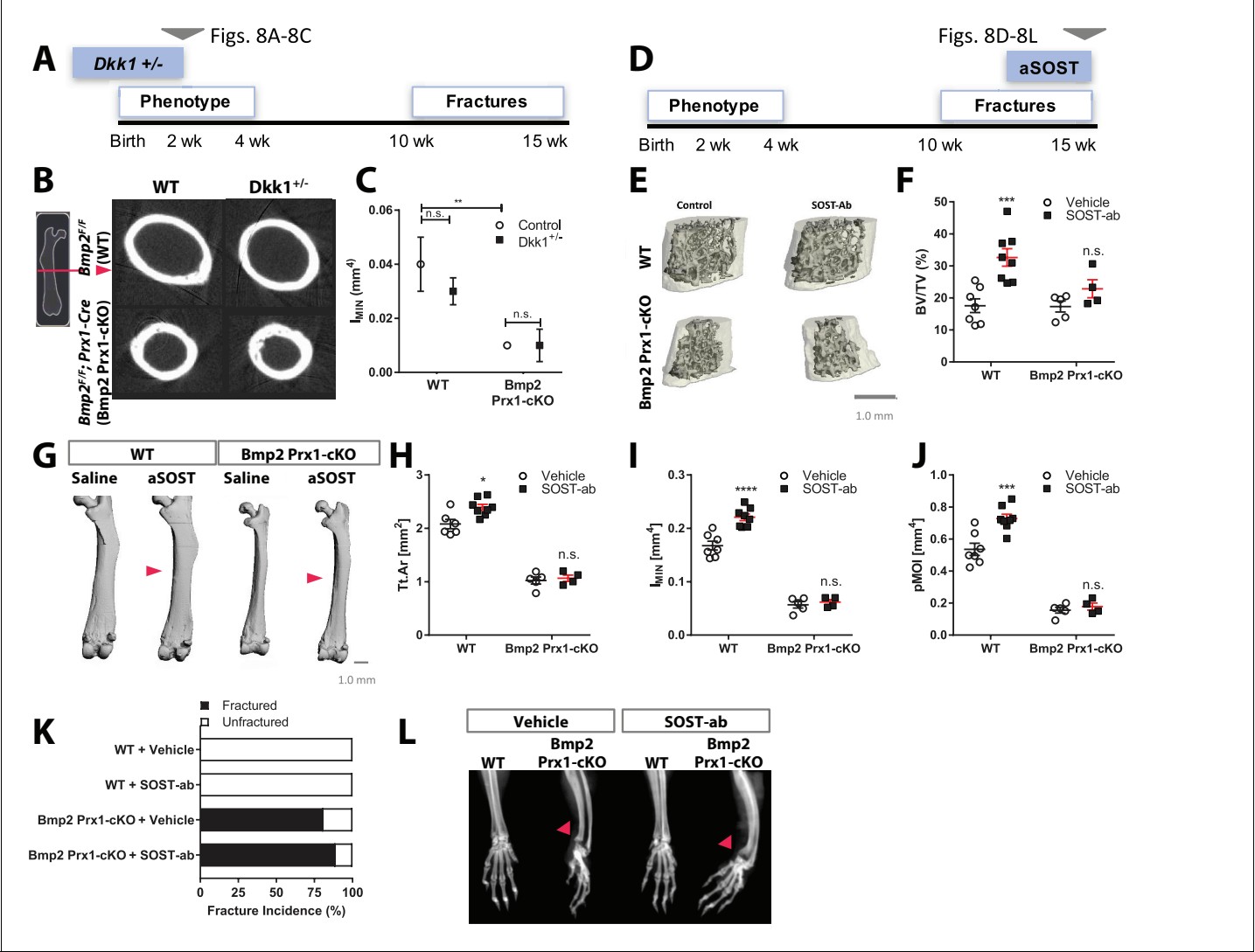

**Figure 8.** *Bmp2* acts downstream of sclerostin neutralizing antibody in the periosteum. (a–c) Haploinsufficiency of *Dkk1* does not rescue periosteal growth in juvenile *Bmp2* Prx1-cKO mice. Femoral bone mass was analyzed by microCT in juvenile mice (two weeks-old). (b) Transverse sections of the femur mid-diaphysis were visualized by 3D reconstructions. Images represent the group mean and are shown to scale. (c) Calculated areal moment of inertia. *n* = 6–8 shown as mean ±s.d. **p<0.005 compared using 2-way ANOVA. (d–l) Pharmacologic activation of Wnt pathway does not rescue periosteal growth or fracture repair in adult *Bmp2* Prx1-cKO mice. WT or *Bmp2* Prx1-cKO mice (13 weeks-old) were treated with sclerostin neutralizing antibody (SOST-ab, 20 mg/kg, two times/week for 2 weeks, subcutaneous). Femoral bone mass was analyzed by microCT. (e) Trabecular bone at the distal metaphysis and (g) cortical bone at the mid-diaphysis of the femur visualized by 3D reconstructions. Images represent the group mean. Scale bars, 1 mm. (f,h–j) Quantitative microCT data presented as mean ±s.d. *p<0.05, ***p<0.005, or ****p<0.00005. vs. matched genotype vehicle control (*n* = 4–8 per group), compared using 2-way ANOVA and Tukey multiple comparisons test. (f) Ratio of bone volume (BV) to trabecular volume (TV) at the distal femoral metaphysis. (h) Total cross-sectional area at the mid-diaphysis. (i) Polar moment of inertia at the mid-diaphysis. (j) Minimum moment of inertia at the mid-diaphysis. (k,l) X-ray imaging revealing non-union forelimb fractures in mice treated two weeks with SOST-ab (*n* = 17–24 wrists per group).

DOI: https://doi.org/10.7554/eLife.42386.018

cells from the mouse embryonic limb bud for the following reasons: early post-natal cortical bone where our phenotype of interest occurs, derives from the embryonic bone collar that forms at the perichondrium at ~E15.5; this bone collar is made by *Sp7+* cells that first emerge at the perichondrium at E13.5 (*Rodda and McMahon, 2006*), subsequently populate the post-natal periosteum and persist for the first month of life (*Maes et al., 2010*); *Sp7+* cells that appear at E13.5 are specified from lateral plate mesoderm of the *Prx1+* limb bud (*Nishimura et al., 2012*; *Martin et al., 1995*); *Prx1+* limb bud first appears at ~E9.5 (*Logan et al., 2002*). The gene regulatory network that

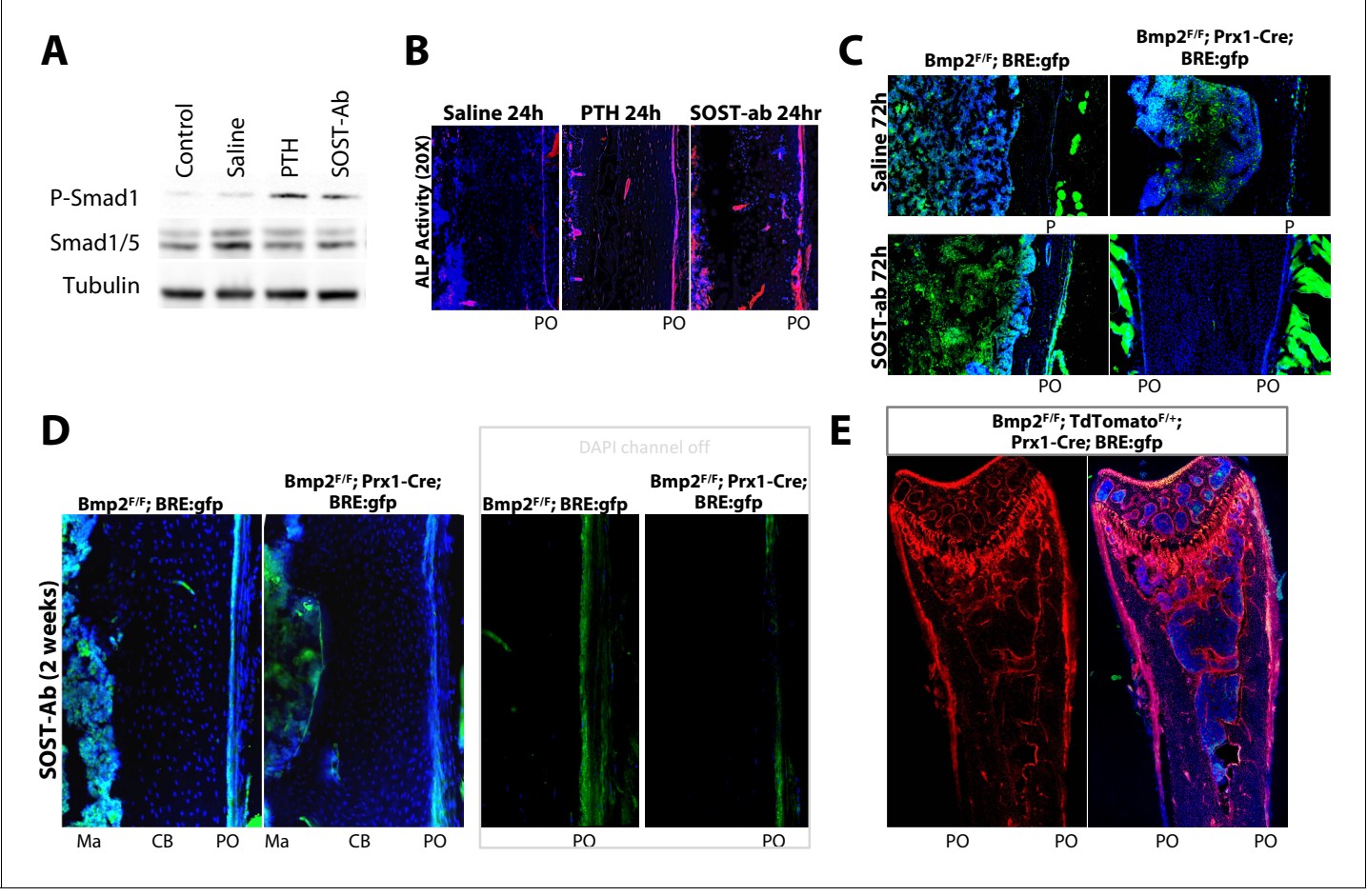

**Figure 9.** *Bmp2* acts downstream of sclerostin neutralizing antibody to reactivate the developmental periosteal BMP signaling center. (a,b) PTH$_{1-34}$ and SOST-ab activate BMP signaling in bone and alkaline phosphatase activity in WT mice. Adult WT mice (4 months-old) were given a single injection of PTH$_{1-34}$ (100 mg/kg, subcutaneous) or SOST-ab (20 mg/kg, subcutaneous). (a) Immunoblot for total and phospho-activated Smads1/5 in marrow-free bone tissue. Experiment was repeated twice. (b) Alkaline phosphatase activity (red) counterstained with DAPI (blue) were imaged on longitudinal sections of the femur (*n* = 2–3 mice per group). (c–e) SOST-ab activates BMP signaling in the periosteum in a *Bmp2*-dependent manner. *BRE:gfp* and a Cre-dependent *tdTomato* reporter were bred onto a *Bmp2* Prx1-cKO background. (c–d) *BRE:gfp* (green) and (e) *tdTomato* (red) was visualized in femurs of adult mice (4 months-old) given 2 injections of (c, top) saline, (c, bottom) 2 injections of SOST-ab (20 mg/kg, subcutaneous) and sacrificed 72 hr after the first injection or (d–e) 4 injections of SOST-ab (20 mg/kg, two times/week for 2 weeks, subcutaneous). Abbreviations: Ma, marrow; CB, cortical bone; PO, periosteum.

DOI: https://doi.org/10.7554/eLife.42386.019

The following figure supplements are available for figure 9:

**Figure supplement 1.** Magnified images of *BRE-gfp* expression in cortical bone following SOST-ab treatment.
DOI: https://doi.org/10.7554/eLife.42386.020

**Figure supplement 2.** *BRE-gfp* expression relative to Prx1-Cre; TdTomato$^{+/Flox}$ lineage, with separated fluorescent channels.
DOI: https://doi.org/10.7554/eLife.42386.021

specifies *Sp7+* cells that make the bone collar and populate the early postnatal periosteum must therefore be highly active in the limb bud between E9.5-E13.5.

In a WNT-ON state, activated canonical WNT transcription factor complexes, comprised of β-catenin and co-factors such as Pygo and Bcl9 (*Städeli and Basler, 2005*), assemble at genomic WNT-responsive elements via interaction with DNA-binding members of the TCF/LEF family. In a WNT-OFF state, TCF/LEF proteins are constitutively bound to chromatin where they function as transcriptional repressors of their bound target genes. Since TCF pulldown is therefore not able to discriminate between genes that are being repressed or induced, we performed anti-Bcl9 ChIP-sequencing (*Cantù et al., 2017*) to define genomic loci targeted by canonical WNT signaling in E10.5 mouse

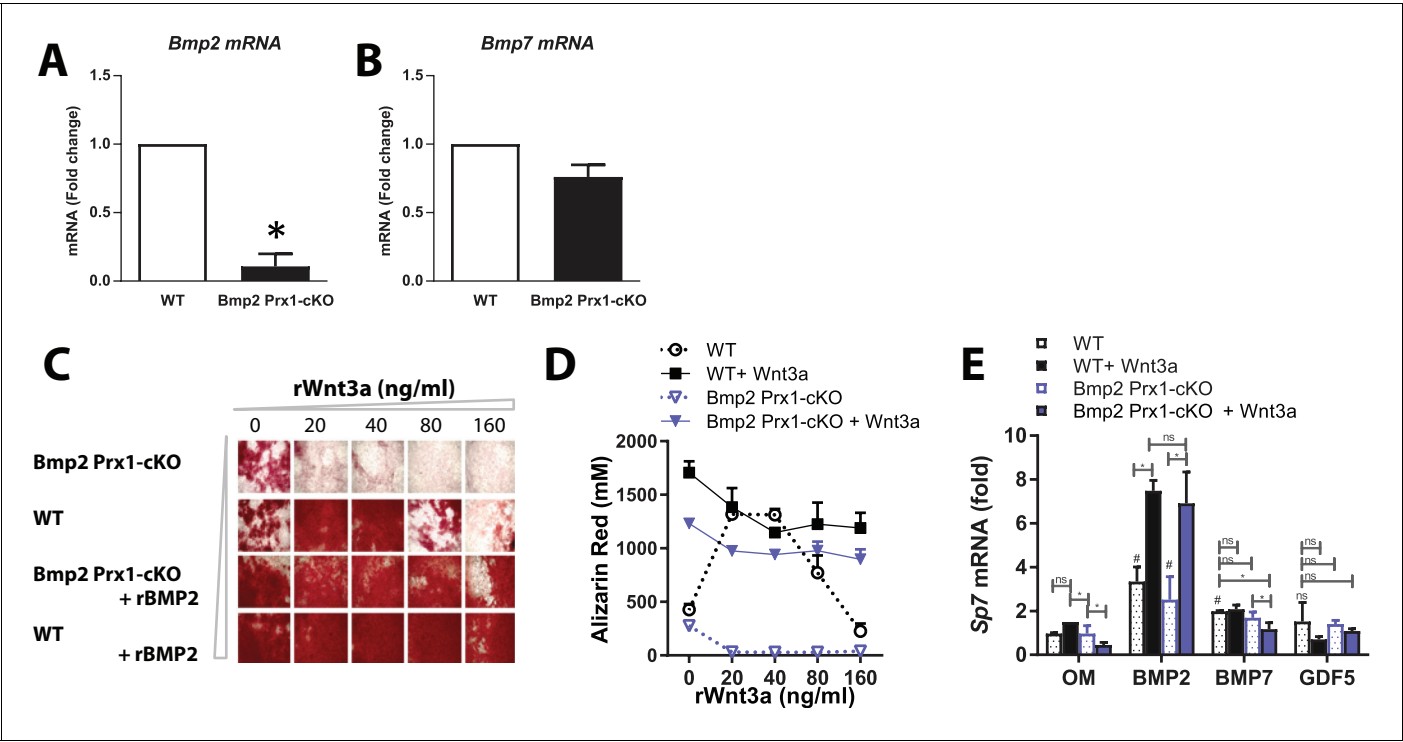

**Figure 10.** Deposition of mineralized bone matrix is optimal when BMP2 and canonical Wnt signaling are balanced. (a,b) Primary periosteal cells isolated from 4 week-old *Bmp2^F/F^* and *Bmp2^F/F^; Prx1-Cre* mice were analyzed by QPCR in three repeat experiments. Fold change mRNA is reported as mean ±s.d. compared by two-tailed student's *t*-test where p*<0.001 vs. *Bmp2^F/F^* cells. *Bmp4* expression was at the limit of detection. Primary BMSC were differentiated in osteogenic medium (OM) plus recombinant growth factors. (c) Calcified matrix was assessed on day 10 by alizarin red staining (2.5X, brightfield). Cultures were performed in duplicate using pooled cell populations from *n* = 2 *Bmp2^F/F^* or *n* = 4 *Bmp2^F/F^; Prx1-Cre* mice. (d) Quantification of alizarin red in (c). Error bars represent distribution of two independent experiments. (e) Primary BMSC cells from *n* = 3 *Bmp2^F/F^* and *n* = 4 *Bmp2^F/F^; Prx1-Cre* mice were differentiated as non-pooled cultures in OM plus recombinant growth factors as indicated. QPCR analysis on day three was reported as mean ±s.d. compared by two-tailed student's *t*-test where *P*\*<0.001 vs. *Bmp2^F/F^* cells in OM; *P*^#^<0.001 vs. *Bmp2^F/F^; Prx1-Cre* cells in OM; *n* = number of independent cultures per condition.

DOI: https://doi.org/10.7554/eLife.42386.022

limb buds (*Figure 11a*). We discovered 2099 high-confidence Bcl9 peaks enriched at gene loci involved in limb, skeletal, and bone morphogenesis (*Figure 11b*) along with previously established canonical WNT target genes to confirm that the strategy had worked (*Figure 11c,d*). KEGG pathway analysis also revealed prominent association of Bcl9 peaks with WNT, TGF-beta, Shh, and Hippo pathways (*Figure 11b*). Importantly, Bcl9 peaks marked multiple regions, including transcriptional start sites, surrounding *Bmp2* and *Grhl3* loci, suggesting direct transcriptional regulation (*Figure 11e,f*). A Bcl9 peak downstream of *Sp7* did not pass statistical threshold. WNT-dependent transcription is therefore already active but does not directly target *Sp7* in the E10.5 limb bud, compatible with the observation that *Sp7+* osteoblasts do not appear until ~E13.5 (*Rodda and McMahon, 2006*). Consistent with the conclusion that *Bmp2* is a WNT target gene, recombinant Wnt3a induced *Bmp2* in limb bud cells (*Figure 11—figure supplement 1a*).

By contrast, chromatin IP qPCR analysis on E13.5 mouse limb bud cells (*Figure 12a*) revealed BMP-dependent recruitment of Smad1 and Grhl3 to a genomic site 13 kb upstream of *Sp7* (*Figure 12b,c*). This site was enriched for active chromatin histone marks and evolutionarily conserved (*Figure 12c*), supporting a cis-regulatory function. In turn, Sp7 is essential for bone formation through direct cis-regulatory control of *Col1A1* (*Hojo et al., 2016*). This major extracellular matrix protein produced by osteoblasts expressed in wild-type but not *Bmp2* Prx1-cKO periosteum (*Figure 5d*).

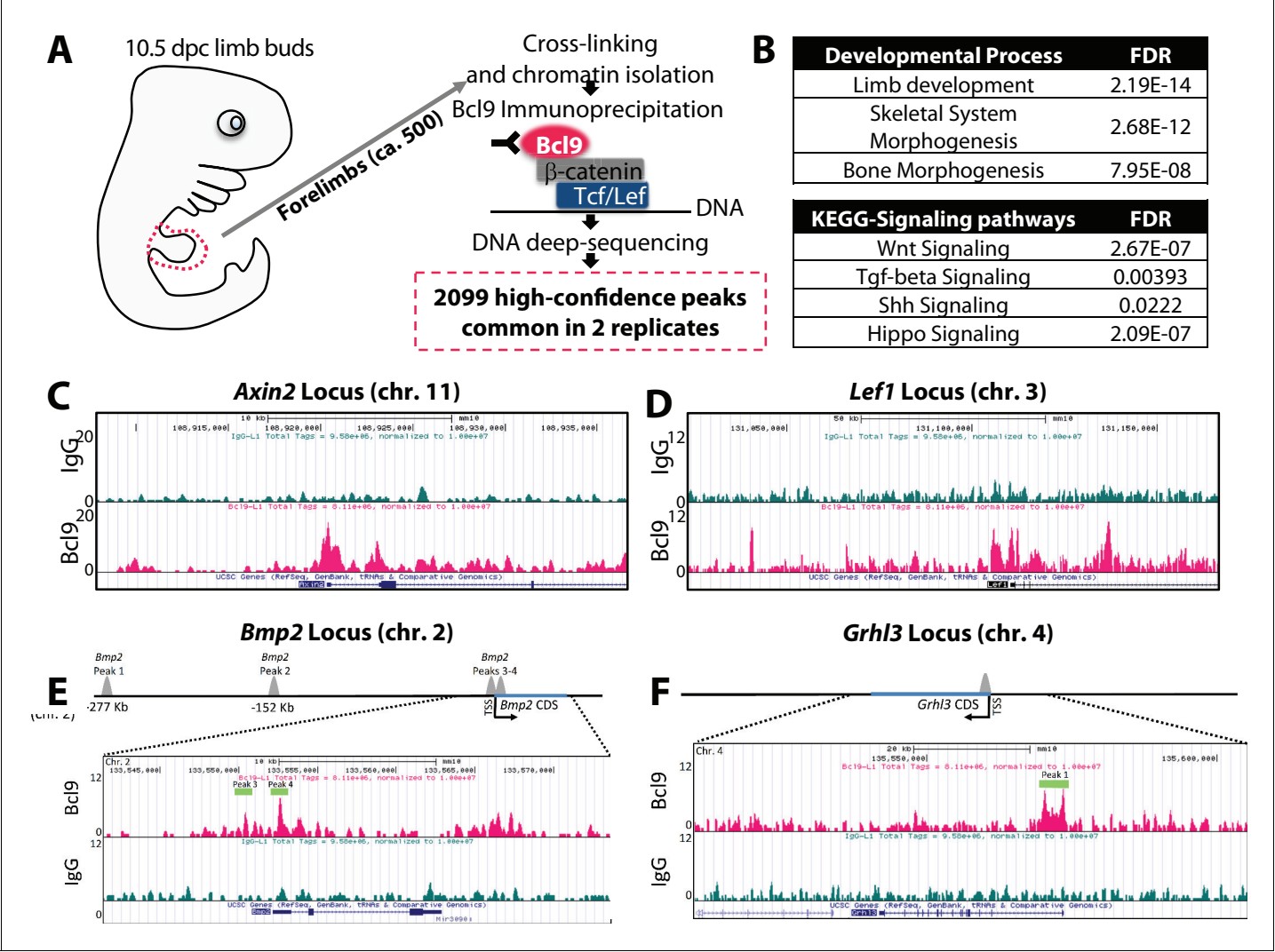

**Figure 11.** *Bmp2* is a direct target gene of canonical Wnt pathway in osteoblast progenitors. (a) Cartoon summarizing discovery of canonical WNT target genes by anti-Bcl9 chromatin immunoprecipitation from E10.5 mouse limb buds, deep sequencing, and bioinformatic analysis of peaks. 2099 high-confidence peaks passed statistical threshold in two independent experiments, with significant enrichment at genetic loci associated with (b) limb, skeletal, and bone development as well as Wnt, TGF-beta, Shh, and Hippo signaling pathways. High-confidence peaks surrounding genetic loci for (c) *Axin2*, (d) *Lef1*, (e) *Bmp2,* and (f) *Grhl3*. Bioinformatic and statistical analysis are described in methods.

DOI: https://doi.org/10.7554/eLife.42386.023

The following figure supplement is available for figure 11:

**Figure supplement 1.** *Bmp2* is upregulated by canonical Wnt pathway in osteoblast progenitors.

DOI: https://doi.org/10.7554/eLife.42386.024

## Human variants of *BMP2* and *GRHL3* are associated with increased risk of fracture

*BMP2* variants predicted to result in haploinsufficiency are associated with short stature, craniofacial gestalt, skeletal anomalies, and congenital heart disease (*Tan et al., 2017*). To further evaluate the clinical consequences of *BMP2* variation, we performed a phenomewide association study of *BMP2* and a downstream effector *GRHL3* in 61,062 individuals from DiscovEHR (http://www.discovehr-share.com/), a cohort linking exome sequence data to electronic health records (EHRs) (*Dewey et al., 2016*). Using a Bonferroni significance threshold of p<1.86e-7 for 268,192 association results, we observed three significant associations for *BMP2* and six significant associations for *GRHL3*. Of note, there was a significant association between a missense variant in *BMP2* (p.

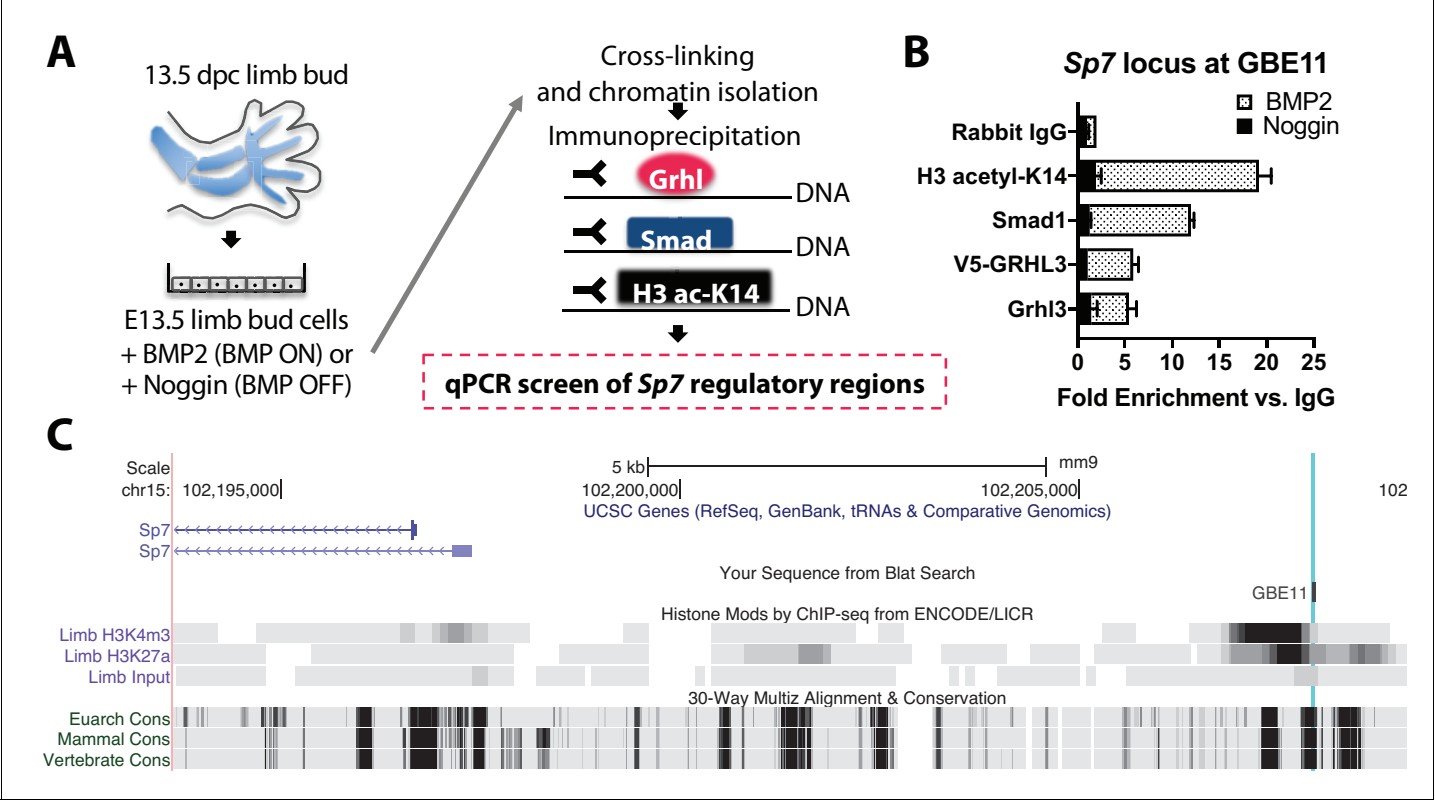

**Figure 12.** *Bmp2* acts downstream of canonical Wnt pathway to specify *Sp7+/Col1A1+* osteoblasts. (a) Cartoon summarizing regulatory analysis of *Sp7* locus by chromatin immunoprecipitation of H3-acetyl-K14, Smad1, endogenous Grhl3, or transiently expressed V5-GRHL3 from chromatin of immortalized E13.5 limb bud cell cultures grown in presence of recombinant BMP2 or Noggin. (b) QPCR analysis revealed BMP-dependent enrichment of these proteins at a putative consensus-binding motif for Grhl3 located at a (c) highly conserved genomic region ~13 kb upstream of the *Sp7* transcription start site. Data reflect mean ±distribution in two independent experiments.

DOI: https://doi.org/10.7554/eLife.42386.025

Arg131Ser) and increased risk of lower leg fracture (odds ratio (OR) 16.05, 95% confidence interval (CI) 6.44–40.00, p=2.54e-9), consistent with previous data on *BMP2* haploinsufficiency and observations in the conditional knockout mouse. Moreover, significant associations were observed between a synonymous variant in *GRHL3* and increased risk of fracture of thoracic vertebra (OR = 12.72, 95% CI 5.59–28.94, p=1.36e-9) and between an intronic variant in *GRHL3* and increased risk of fracture of patella (OR = 13.33, 95% CI 5.11–34.81, p=1.22e-7) (*Table 5*).

In summary, our data demonstrate that skeletal development, fracture repair, and therapeutic response to systemic treatment with PTH or SOST-ab all converge on expression of *Bmp2* in the periosteal niche for control of osteoblast specification and periosteal function (*Figure 13a*).

## Discussion

Whereas the signaling mechanisms controlling longitudinal growth by the growth plate were initially described in 1996 (*Lanske et al., 1996*; *Vortkamp et al., 1996*), the mechanisms underlying periosteal growth by the periosteum, the primary determinant of bone width and therefore bone strength, have remained unknown. Here we reveal that *Bmp2* governs all thus-far identified developmental, clinically-inducible and repair functions of the periosteum. *Bmp2* is dispensable for periosteum formation, but essential for osteoblast differentiation from periosteal progenitors, and thereby periosteal bone growth and fracture repair. Local BMP2 expression regulates BMP signaling in the periosteal niche during this process and drives periosteal response to canonical WNT and PTH signals.

Global loss of *Bmp2* is embryonic lethal, explaining why the identity of a signal controlling periosteal bone growth, a postnatal function, eluded us for so long. The slender bone phenotype in *Bmp2* Prx1-cKO mice, established by 2 weeks of age, is not recapitulated by omission of any other BMP from bone (*Salazar et al., 2016a*), nor is it present when *Bmp2* is deleted in more specified skeletal progenitors expressing *Sp7-EGFP::Cre* or *2.3kb-Col1a1-Cre*.

Our work reveals the innermost layer of periosteum as a robust BMP signaling center and identifies pre-*Sp7* +progenitors as a critical source and target for BMP2. Cells in the outer periosteum and within cortical bone are not highly engaged in BMP signaling. BMP2 availability in bone therefore appears to be tightly regulated, perhaps through sequestration of BMP2 in the extra-cellular matrix. Comparative analysis of *Bmp2^{lacz}* with *BRE:gfp* strongly suggests that cortical bone matrix functions as a BMP2 repository, allowing only local activation of cells at the innermost cambial periosteal layer. Future studies uncovering the mechanisms that restrict BMP2 bioavailability to the periosteum could uncover new strategies to improve bone width, resistance to fracture, and bone repair.

Excessive LRP5/6 signaling underlies several human skeletal disorders characterized in part by exuberant periosteal expansion (*Baron and Kneissel, 2013*). Strikingly however, periosteal growth and fracture repair defects in *Bmp2 cKO* mice were not rescued by haploinsufficiency of *Dkk1*, iPTH therapy, or treatment with SOST-ab. We propose that LRP5/6-dependent pathways initiating periosteal bone formation in the postnatal skeleton converge on upregulation of *Bmp2* for induction of BMP signaling. BMP2 signaling, by coordinating assembly of a Smad/Grhl3 complex at the *Sp7* genomic locus, thereby acts downstream of canonical WNT to specify the *Sp7+/Col1A1+* cells required for periosteal growth and fracture repair.

Looking forward, we find a lack of information and understanding of the signals mediating normal BMP2 expression in the periosteal niche, both as a function of age and in the context of skeletal disease. This information is key to modulating skeletal stem cell behavior to improve healing and in response to therapeutic agents. As all of our experiments are done in the total absence of periosteal BMP2, the information needed to figure out this part of the puzzle remains elusive, especially since deposition of BMP2 in bone and new synthesis of BMP2 are both missing in our mice. An abundance of cis-regulatory elements in the BMP2 gene desert mediate precise spatiotemporal control of BMP2 during development and repair, and this could potentially be exploited for screens identifying improved and more controlled methods of bone repair mediated by endogenous expression of *Bmp2*.

In summary, the dynamic spatiotemporal expression pattern of *Bmp2* constitutes an essential mechanism determining active versus quiescent states of the periosteal niche throughout embryonic and postnatal life. We identify *Bmp2* as the signal that couples bone length to bone width during development and demonstrate that reactivation of this developmental signaling center is the essential mechanism by which bone anabolic therapies recruit the periosteum to reduce fracture risk and accelerate fracture repair.

## Materials and methods

**Key resources table**

| Reagent type (species) or resource | Designation | Source or reference | Identifiers | Additional information |
|---|---|---|---|---|
| Gene (*Mus musculus*) | bone morphogenetic protein 2 (*Bmp2*) | | MGI:MGI:88177 | |
| Genetic reagent (*Mus musculus*) | Tg(Prrx1-cre)1Cjt; Bmp2tm1Cjt/Bmp 2tm1Cjt | PMID:17194222 | MGI:3700047; RRID:MGI:3700047 | |
| Genetic reagent (*Mus musculus*) | B6.Cg-Tg (Sp7-tTA,tetO-EGFP/cre)1Amc/J | PMID:16854976 | IMSR JAX:006361; RRID:IMSR_JAX:006361 | |
| Genetic reagent (*Mus musculus*) | B6.FVB-Tg(Col1a1-cre) 1Kry/Rbrc | PMID:12112477 | IMSR:RBRC05603; RRID:IMSR_RBRC05603 | |

*Continued on next page*

*Continued*

| Reagent type (species) or resource | Designation | Source or reference | Identifiers | Additional information |
|---|---|---|---|---|
| Genetic reagent (*Mus musculus*) | Bmp2tm1(KOMP)Vlcg/Bmp2+ | PMID:29198724 | MGI:5912401; RRID:MGI:5912401 | |
| Genetic reagent (*Mus musculus*) | BRE:gfp | PMID:18615729 | | |
| Genetic reagent (*Mus musculus*) | Dkk1tm1Lmgd/Dkk1tm1Lmgd | PMID:17127040; PMID:11702953 | MGI:3618757; RRID:MGI:3618757 | |
| Genetic reagent (*Mus musculus*) | B6.Cg-Gt(ROSA)26Sortm9(CAG-tdTomato)Hze/J | Jackson Laboratory | IMSR JAX:007909; RRID:IMSR_JAX:007909 | |
| Cell line (*Mus musculus*) | MLB13 Clone 14 | PMID:7532346; PMID:8302904 | | |
| Transfected construct (*H. sapien*) | V5-tagged GRHL3 | Center for Cancer Systems Biology | PlasmID_clone: HsCD00376192 | |
| Antibody | anti-Id1 (rabbit polyclonal) | Santa Cruz | Santa Cruz Biotechnology:sc-488; RRID:AB_631701 | Immunostaining (1:100) |
| Antibody | anti-Id3 (mouse polyclonal) | Santa Cruz | Santa Cruz Biotechnology:sc-490; RRID:AB_2123010 | Immunostaining (1:100) |
| Antibody | anti-pSmad1/5/8 (rabbit monoclonal) | Cell Signaling Technologies | Cell Signaling Technology:9511; RRID:AB_331671 | Western (1:1000); Immunostaining (1:50) |
| Antibody | anti-Smad1 (rabbit polyclonal) | Cell Signaling Technologies | Cell Signaling Technology:9743; RRID:AB_2107780 | Western (1:1000); ChIP (1:25) |
| Antibody | anti-alpha-Tubulin (mouse monoclonal) | Sigma-Aldrich | Sigma-Aldrich:T6074; RRID:AB_477582 | Western (1:1000) |
| Antibody | anti-V5-tag (mouse monoclonal) | Abcam | Abcam Cat:ab27671; RRID:AB_471093 | ChIP (1:250) |
| Antibody | anti-GRHL3 (rabbit polyclonal) | Thermo Fisher | Thermo Fisher Scientific:PA5-41616; RRID:AB_2606412 | ChIP (1:100) |
| Antibody | anti-H3acK14 (rabbit polyclonal) | Millipore | Millipore:06–599; RRID:AB_2115283 | ChIP (1:100) |
| Antibody | anti-Bcl9 (rabbit polyclonal) | Abcam | Abcam:ab37305; RRID:AB_2227890 | ChIP (1 μg) |
| Software, algorithm | Biotapestry | Biotapestry.org; PMID:15907831; PMID:18757046; PMID:27134726 | | |

## Animals

In vivo experiments were performed in compliance with the Guide for the Care and Use of Laboratory Animals and were approved by the Harvard Medical Area Institutional Animal Care and Use Committee (protocol #04043 to V.R.). Mice bearing alleles where loxP sites flank the coding sequence of exon 3 of *Bmp2* (*Bmp2^F/F^*) were bred to *Prx1-Cre* mice (*B6.Cg-Tg(Prrx1-cre)1Cjt/J*) (*Logan et al., 2002*), *Sp7-GFP::Cre* mice (B6.Cg-Tg(Sp7-tTA,tetO-EGFP/cre)1Amc/J) (*Rodda and McMahon, 2006*), or 2.3kb-*Col1A1-Cre* mice (*Tg(Col1a1-cre)1Kry*) (*Dacquin et al., 2002*) to obtain *Bmp2^Flox/Flox^*; *Prx1-Cre* mice (*Bmp2* Prx1-cKO), *Bmp2^Flox/Flox^*; *Sp7-EGFP::Cre* mice (*Bmp2* Sp7-cKO), or *Bmp2^Flox/Flox^*; *Col1a1-Cre* mice (*Bmp2* Col1a1-cKO). Mice carrying floxed *Bmp2* alleles (*Bmp2^Flox/Flox^*) were crossed to *Dkk1* haploinsufficient mice (*Dkk1tm1Lmgd/Dkk1tm1Lmgd*)

**Table 5.** Human variants of *BMP2* and *GRHL3* are associated with increased risk of fractures.

We performed a phenomewide association study of *BMP2* and a downstream effector *GRHL3* in 61,062 individuals from DiscovEHR, a cohort linking exome sequence data to electronic health records (EHRs). Using a Bonferroni significance threshold of p<1.86e-7 for 268,192 association results, we observed three significant associations for *BMP2* and six significant associations for *GRHL3*.

| Variant | Gene | Functional Prediction | HGVS amino acid | Phenotype | Odds Ratio (CI) | P-Value | MAF |
|---|---|---|---|---|---|---|---|
| 20:6770235: T:G | *BMP2* | missense | p.Ser37Ala | Post-eruptive color changes of dental hard tissues | 15.44 (6.50–36.64) | 5.33E-10 | 0.01802 |
| 20:6778359: G:A | *BMP2* | missense | p.Arg154Gln | Secondary hyperparathyroidism, not elsewhere classified | 18.88 (7.35–48.53) | 1.06E-09 | 0.00033 |
| 1:24336821: A:T | *GRHL3* | synonymous | p.Pro202Pro:p.Pro156Pro:p.Pro207Pro: p.Pro202Pro | Fracture of thoracic vertebra | 12.72 (5.59–28.94) | 1.36E-09 | 0.00078 |
| 20:6778291: A:T | *BMP2* | missense | p.Arg131Ser | Fracture of lower leg, including ankle | 16.05 (6.44–40.00) | 2.54E-09 | 0.00019 |
| 1:24342967: C:T | *GRHL3* | missense | p.Thr454Met:p.Thr408Met:p.Thr459Met: p.Thr454Met | Atresia of bile ducts | 17.11 (6.34–46.18) | 2.07E-08 | 0.03395 |
| 1:24336694: T:C | *GRHL3* | missense | p.Val160Ala:p.Val114Ala:p.Val165Ala:p. Val160Ala | Malignant neoplasm of parietal lobe | 8.72 (4.08–18.64) | 2.31E-08 | 0.04732 |
| 1:24344928: A:G | *GRHL3* | missense | p.Asn484Ser:p.Asn438Ser:p.Asn489Ser: p.Asn484Ser | Malignant neoplasm of parietal lobe | 8.23 (3.90–17.38) | 3.24E-08 | 0.0484 |
| 1:24339645: C:G | *GRHL3* | intronic | NA | Fracture of patella | 13.33 (5.11–34.81) | 1.22E-07 | 0.00109 |
| 1:24342967: C:T | *GRHL3* | missense | p.Thr454Met:p.Thr408Met:p.Thr459Met: p.Thr454Met | Atresia of esophagus with tracheo-esophageal fistula | 9.76 (4.16–22.88) | 1.63E-07 | 0.03396 |

DOI: https://doi.org/10.7554/eLife.42386.026

(*Mukhopadhyay et al., 2001*) to obtain *Dkk1$^{+/-}$; Bmp2$^{Flox/Flox}$* mice, and these mice were subsequently bred to *Bmp2$^{+/Flox}$; Prx1-Cre* mice to obtain *Dkk1$^{+/-}$; Bmp2$^{Flox/Flox}$; Prx1-Cre* mice (*Dkk1$^{+/-}$; Bmp2* Prx1-cKO). *Bmp2* Prx1-cKO mice were also crossed to Ai9 Cre-reporter mice (*B6.Cg-Gt (ROSA)26Sortm9(CAG-tdTomato)Hze/J*) harboring a loxP-flanked STOP cassette preventing transcription of a CAG promoter-driven red fluorescent protein variant (*tdTomato*). The targeted mutation was inserted into the *Gt(ROSA)26Sor* locus by homologous recombination. TdTomato is expressed when bred to mice that express Cre recombinase.

Mice with a *lacZ* cassette knock-in to the endogenous *Bmp2* locus (*Bmp2$^{tm1(KOMP)Vlcg/+}$*), herein referred to as *Bmp2$^{lacz}$*) were generated by the trans-NIH Knock Out Mouse Project (KOMP) and obtained from the KOMP repository (www.komp.org). The genomic region of the *Bmp2* containing exon two was deleted and replaced with a transmembrane-lacZ/neo cassette using bacterial homologous recombination. Transgenic mice expressing enhanced green fluorescent protein under the control of a BMP-responsive fragment of the *Id1* promoter (herein referred to as *BRE:gfp*) (*Monteiro et al., 2008*) were a kind gift of Dr. Christine Mummery.

## Bone measurements

Femoral length and width (antero/posterior) were measured in fresh collected bones using a digital caliper (Ted Pella, Inc) (8–20 per age and per genotype). Percentage of tissue area and cross-sectional composition (*Figure 1*) were calculated based on areal measurements obtained with ImageJ Software areal measurement tool. The whole radius from newborn (n = 6–9) and P14 (n = 3–9) *Bmp2* Prx1-cKO mice and control littermates were sectioned from distal to proximal. One section every 50

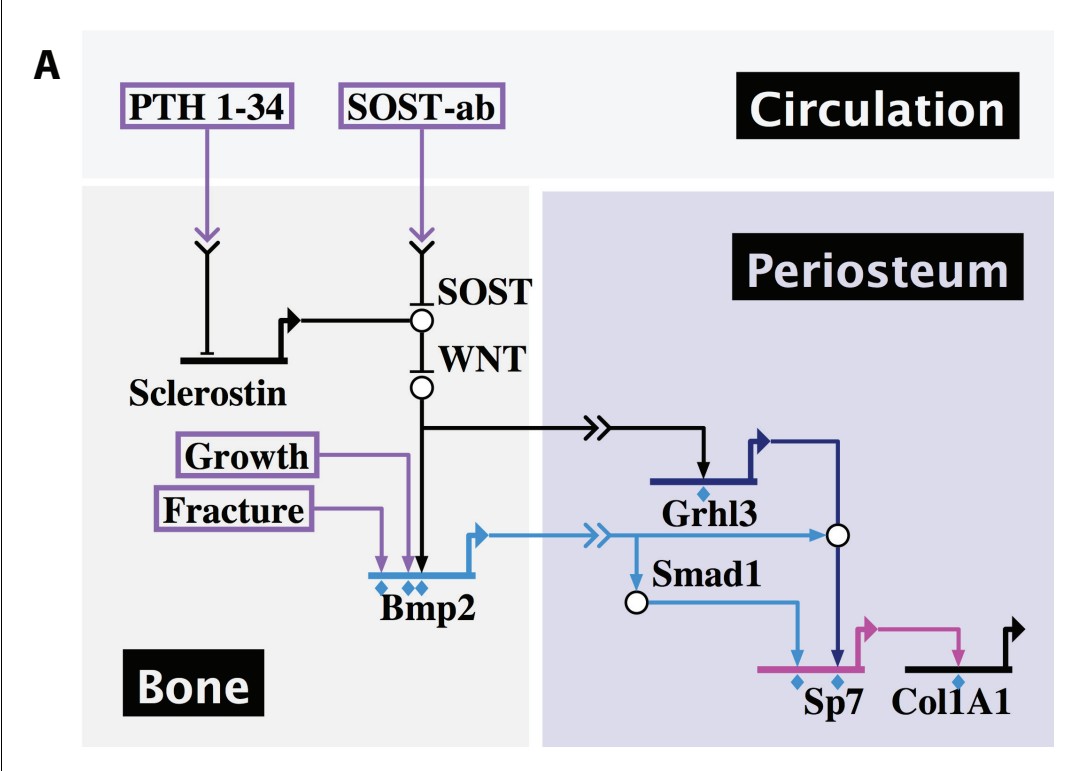

**Figure 13.** Developmental, reparative, and therapeutic signals converge on *Bmp2* to specify osteoblasts in the periosteal niche. (a) A proposed hierarchical and regionalized gene regulatory network, summarizing the source and identity of signals that regulate transcription of *Bmp2* for downstream specification of *Sp7+/Col1A1+* osteoblasts in the periosteal niche. Boxed regions, biological compartments; Boxed text, inputs evaluated in this study; Double arrows in a link, communication across territories; Flat footpads ending at genetic loci, transcriptional inhibition; Pointed footpads ending at genetic loci, transcriptional induction; Diamonds under footpads, evidence of direct cis-regulatory interaction provided by our study; White bubbles, protein/protein interactions. Made with Biotapestry.org.

DOI: https://doi.org/10.7554/eLife.42386.027

μm was measured. Digital images were obtained using a Zeiss AxioImager MI Microscope fitted with an AxioCam HRC digital camera and Zeiss AxioVision imaging software.

### Radiographs

Imaging was done post-mortem in a Bruker MS-Fx-Pro using 45 kVp of radiation energy and a 30 s exposure time.

### Micro-CT and bone strength assessment

Femora were scanned post-mortem with a μCT40 (Scanco Medical AG, Switzerland). The mid-shaft of the diaphysis was analyzed using a 12 μm voxel size, 70 kVp/145 mA of radiation energy and a 300 msec integration time. Contouring was performed using a threshold of 220 for trabecular bone (450.7 mgHA/cm$^3$) or 334 for cortical bone (787.7 mgHA/cm$^3$). We quantified the average moment of inertia ($I_{MIN}$), and the average total cross-sectional area of bone tissue (Ct.Ar). To assess structural strength, hydrated femurs were loaded in three-point bending to determine the peak moment (maximum force x span) endured by each bone (*Makowski et al., 2014*). For the estimated material properties reported in supplemental materials, we used standard flexural formula (maximum force x span x $C_{MIN}/I_{MIN}$) from beam theory the structural properties from microCT. Data are reported as mean ±s.d, where p-value was calculated using 1-way ANOVA and post-test Newman Keulus. All data points were included in the analysis.

## Intermittent parathyroid hormone

Two week-old *Bmp2* Prx1-cKO mice and control littermates were assigned to treatment groups (*n* = 5–8 per group). PTH$_{1\text{-}34}$ (hPTH(1–34); Bachem, Bubendorf, Switzerland) was administered at 100 µg/kg/day (*Kramer et al., 2010*) for 2 weeks by subcutaneous injections or equivalent volume of vehicle (phosphate-buffered saline, pH 7.4) seven days per week. Seven week-old male *Bmp2* Prx1-cKO mice and control littermates were treated for 3 weeks with PTH$_{1\text{-}34}$ or vehicle five days per week (*n* = 4–8). Mice were sacrificed 2 hr after the final PTH injection, followed by exsanguination and tissue collection for further analysis.

## Histomorphometry

Static and dynamic histomorphometry measurements were analyzed between 4 week-old or 10 week-old *Bmp2* Prx1-cKO mice and control littermates treated or not treated with hPTH. In the first set of experiments (2 to 4 weeks of age) mice were injected with demeclocycline hydrochloride (40 mg/kg) for 4 days before euthanasia and with calcein (20 mg/kg) 1 day before euthanasia (*Fonseca et al., 2011*). In the second set of experiments (7 to 10 weeks of age) the same doses of demeclocycline and calcein were administered, but 8 and 2 days before euthanasia (*Guo et al., 2010*). Kinetic bone parameters were obtained from unstained 10 µm sections examined by fluorescent light microscopy. The static and dynamic histomorphometric indexes were evaluated in trabecular and cortical bone in sagittal and transversal sections. Periosteal and endosteal parameters were evaluated as described previously (*Millard et al., 2011*).

## Sclerostin-neutralizing antibody

SOST-ab (provided by A. Economides at Regeneron) was prepared in sterile saline and delivered via subcutaneous injection at a dose of 20 mg/kg, twice per week for 2 weeks. Diluted antibody was stored in single-use aliquots at −80°C.

## Histology

Samples removed for histology were fixed in 4% paraformaldehyde, decalcified in Tris buffer containing 10% EDTA and embedded in paraffin. Sections (5 mm thick) were stained with 0.1% toluidine blue using standard procedures. Immunolocalization was performed as described (*Retting et al., 2009*; *Lowery et al., 2010*). Briefly, sections were digested with trypsin (1% in PBS) for 10 min at 37°C and then boiled for 15 min in citrate buffer (*Ivkovic et al., 2003*). Sections were blocked with 5% goat serum for 1 hr and incubated with primary antibody overnight at 4°C, followed by incubation with secondary antibody for 1 hr at room temperature, then with fluorophore for 30 min at room temperature or incubated with Vector Elite ABC reagent (Vector) for 60 min before developing with diaminobenzidine. Primary antibodies were as follows: anti-pSMADS (Cell Signaling); anti-ID1 and ID3 (BioCheck); anti-IGF1 (Abcam). Secondary antibodies were conjugated with AlexaFluor-555 and AlexaFluor-488 for immunofluorescence and sections were counterstained with DAPI (Vectashield). Secondary antibodies were conjugated with biotin for immunohistochemistry and sections were counterstained with toluidine blue or methyl green. Digital images were obtained using a Zeiss AxioImager MI Microscope fitted with an AxioCam HRC digital camera and Zeiss AxioVision imaging software.

## Standardized fractures

with pin stabilization were made in femora of adult mice as previously described (*Tsuji et al., 2006*). Fractured and non-fractured contralateral controls femurs were collected 3 days after fracture, processed for paraffin sectioning, and subjected to lacZ staining.

## LacZ staining

Tissues were embedded in optimal cutting temperature compound (OCT) on dry ice and frozen sections were prepared for beta-galactosidase staining as previously described (*Kokabu et al., 2012*).

## Cryohistology for endogenously expressed reporters

Tissues were fixed in 10% neutral buffered formalin overnight at room temperature. If necessary, tissues were decalcified in 14% EDTA pH 8.0 prior cryopreservation in a 30% sucrose gradient, then

embedded in optimal cutting temperature compound (OCT) on dry ice. Frozen sections from adult bones were collected on a CryoJane tape system and mounted with Vectashield plus DAPI (Vector Labs).

### *Col1A1* in situ hybridization

P14 mouse hindlimbs were fixed, paraffin processed and sectioned in the coronal plane using standard methods. In vitro transcription to generate riboprobes was performed using standard protocols and reagents (Promega). In situ hybridization with digoxigenin-labeled *Col1A1* probe was carried out as described previously (*Gamer et al., 2009*).

### Three-point bending

Femurs were placed on the lower support points of a three-point bending fixture with the anterior side down (*i.e.*, bending about the medial-lateral plane). The span (*L*) between the lower supports was 7 mm or 8 mm depending on genotype. Hydrated bone was loaded-to-failure at 3.0 mm/min (Dynamight 8841, Instron, Canton, OH) to generate the force vs. displacement (P-d) curve. The primary structural properties were rigidity ($\delta$), which is the slope of the linear portion of the curve multiplied by *L* (*Ornitz and Legeai-Mallet, 2017*)/48, ultimate moment ($P_u \times L/4$), which is the peak bending moment endured by the mid-shaft, the post-yield deflection (PYD), which is the normalized displacement at failure minus the displacement at yielding, and work-to-fracture ($W_f$), which is the area under the normalized P-d curve generated by the test (i.e., multiplied by $12/L2$ to adjust for differences in span).

### Serology

Blood was collected by heart puncture after euthanasia for serum biochemistry. Serum IGF-1 was measured with ELISA kit (Immuno diagnostics, Inc, CA, USA) from 2-, 4- and 10-week-old mice. Analysis was performed as suggested by the manufacturer and, in all cases, serum was diluted 1:10.

### Recombinant growth factors

Recombinant human BMP2 (200 ng/ml; Genetics Institute). Recombinant human BMP7 and GDF5 (200 ng/ml; R and D Systems). Recombinant mouse Wnt3a, Wnt10 (40 ng/ml; R and D Systems). Noggin (10 ug/ml; R and D Systems).

### Immunoblot

Bones were cleaned of soft tissue and marrow, snap frozen in liquid nitrogen, and pulverized for 10 min using a Bullet Blender and Navy Lysis Beads in RIPA buffer containing Halt protease and phosphatase inhibitors (Pierce). Proteins were separated by SDS-page electrophoresis and immunoblotted with the following antibodies at 1:1000 dilutions: phospho-Smad1/5 (Cell Signaling #9511), Smad1 (Cell Signaling # 9743), and alpha-Tubulin (Sigma, #T6074).

### Bone marrow stromal cells (BMSC)

Bone marrow from 4 to 6 month-old mice (male and female) was collected as previously described (*Salazar et al., 2013*) and plated in BMSC medium (ascorbic acid free $\alpha$-MEM (Invitrogen) containing 20% FBS, 40 mM L-glutamine, 100 U/ml penicillin-G, and 100 mg/ml streptomycin). After 3 days, non-adherent cells were removed by vigorous washing. Osteoblast differentiation BMSC were seeded in BMSC medium at 35,000 cells/well in 96-well dishes (for alizarin red staining) or 300,000 cells per well in 24-well dishes (for RNA or protein). Confluent cultures were stimulated on day one with Osteogenic Medium (OM: BMSC medium plus 50 µg/ml ascorbic acid and 10 mM $\beta$-glycerophosphate)±200 ng/ml rhBMP2 (Genetics Institute) or 40 ng/ml recombinant mouse Wnt3a (R and D Systems). **Alizarin red** Fixed cells were stained for 30 mins in 0.4% aqueous solution of alizarin red S (Sigma). ARS was eluted in 10% glacial acetic acid, pH was adjusted with 10% ammonium hydroxide, and absorbance was measured at 405 nm. **QPCR** Cells were scraped into 250 µl/well of Trizol, homogenized five times through a 22 g needle and five times through a 25 g needle. Genomic DNA was digested with 50 ul gDNase solution (RNeasy Plus Universal Kit, Qiagen). The organic phase was separated using 50 µl chloroform, and mixed with 150 µl of 70% ethanol before loading onto an RNeasy spin column and purifying according to manufacturer's instructions. RNA was reverse

transcribed with EcoDry Premix (Clonetech). Data were normalized to *beta-actin*, analyzed using ∆∆CT method, and expressed as mean ±s.d relative to *Bmp2$^{F/F}$* cells cultured in osteogenic medium. Two-tailed student's *t*-test was utilized to calculate *P*-values.

## Periosteal cells

Periosteal cells were isolated from hindlimbs and forelimbs of 4 week-old *BMP2$^{F/F}$; Prx1-Cre* mice. Litter mate *BMP2$^{F/F}$* or *BMP2$^{F/+}$: Prx1-Cre* were used as controls. In brief, long bone were carefully cleaned from muscles and the ends of the bones sealed with agarose. The bones were digested with collagenase and dispase 3 mg/ml (Sigma) for 45 min in a shaking 37°C water bath. Cells were collected, strained and plated in 5% CO2% and 5% O2 incubator. After expansion RNA was extracted. Cells from *BMP2$^{F/F}$; Prx1-Cre* mice do not grow well, so cultures were supplemented with 10 ng/ml rBMP2. For each isolation bones from 3 to 4 mice/genotype were pooled to create one culture. Three separate isolations were performed for gene expression analysis. 500 ng RNA was reverse transcribed to cDNA using the iSCRIPT kit (BioRad) and qPCR analysis performed (SYBR GREEN). The data presented as mean ± SEM. BMP4 was detected only in one of the samples.

## Chromatin immunoprecipitation

Immortalized and clonal limb bud cells from E13.5 mouse embryos (MLB13 clone 14) (*Rosen et al., 1993*; *Rosen et al., 1994*) were cultured and maintained in DMEM supplemented with 10% fetal bovine serum and Pen/Strep. MLB13 clone 14 cells were generated by Vicki Rosen and maintained continuously in Dr. Rosen's private storage. They remain neomycin-resistant (based on the strategy for immortalization) and maintain historical features of morphology and differentiation capacity in skeletal lineages. They are free of mycoplasma, as determined by QPCR testing. A plasmid encoding V5-tagged GRHL3 was provided by Center for Cancer Systems Biology, PlasmID clone HsCD00376192. MLB13 were seeded to 90% confluence, transfected with a plasmid encoding human GRHL3, and cultured for 3 d in osteogenic medium plus indicated growth factors. Chromatin immunoprecipitation was performed according to manufacturer's instructions with the SimpleChIP Enzymatic Chromatin IP Kit with magnetic beads (Cell Signaling). Primers for Grhl3 binding element 11 are: Forward 5'-ACCTGGGTATTGCCTGAAAA-3' and Reverse 5'-GGAAGAGCTGGCTTCTTTGA-3'. V5-antibody: Invitrogen #R961-25. GRHL3 antibody: ThermoFisher #PA5-41616. H3acK14 antibody: Millipore #06–599. Rabbit IgG: Cell Signaling #2729. Smad1 antibody: Cell Signaling #9743.

## Chromatin immunoprecipitation/sequencing

Forelimbs buds were manually dissected from ca. 250 RjOrl:SWISS outbred 10.5 dpc mouse embryos. Chromatin immunoprecipitation was performed as previously described (*Cantù et al., 2013*). Briefly, the tissue was dissociated to a single cell suspension with collagenase (1 ug/ml in PBS) for 1 hr at 37° C, washed and crosslinked in 20 ml PBS for 40 min with the addition of 1.5 mM ethylene glycol-bis(succinimidyl succinate) (Thermo Scientific, Waltham, MA, USA), for protein-protein crosslinking (*Schuijers et al., 2014*), and 1% formaldehyde for the last 20 min of incubation, to preserve DNA-protein interactions. The reaction was blocked with glycine and the cells were subsequently lysed in 1 ml HEPES buffer (0.3% SDS, 1% Triton-X 100, 0.15 M NaCl, 1 mM EDTA, 0.5 mM EGTA, 20 mM HEPES). Chromatin was sheared using Covaris S2 (Covaris, Woburn, MA, USA) for 8 min with the following set up: duty cycle: max, intensity: max, cycles/burst: max, mode: Power Tracking. The sonicated chromatin was diluted to 0.15% SDS and incubated overnight at 4°C with 1 µg of anti-Bcl9 (Abcam, ab37305) or IgG and 50 ul of protein A/G magnetic beads (Upstate). The beads were washed at 4°C with wash buffer 1 (0.1% SDS, 0.1% deoxycholate, 1% Triton X-100, 0.15 M NaCl, 1 mM EDTA, 0.5 mM EGTA, 20 mM HEPES), wash buffer 2 (0.1% SDS, 0.1% sodium deoxycholate, 1% Triton X-100, 0.5 M NaCl, 1 mM EDTA, 0.5 mM EGTA, 20 mM HEPES), wash buffer 3 (0.25 M LiCl, 0.5% sodium deoxycholate, 0.5% NP-40, 1 mM EDTA, 0.5 mM EGTA, 20 mM HEPES), and finally twice with Tris EDTA buffer. The chromatin was eluted with 1% SDS, 0.1 M NaHCO3, decrosslinked by incubation at 65°C for 5 hr with 200 mM NaCl, extracted with phenol-chloroform, and ethanol precipitated. The immunoprecipitated DNA was used as input material for DNA deep sequencing. ChIP-seq data are available at Array Express accession number E-MTAB-7652.

## Peak calling

All sequenced reads were mapped using the tool for fast and sensitive reads alignment, Bowtie 2 (http://bowtie-bio.sourceforge.net/bowtie2/index.shtml), onto the UCSC mm10 reference mouse genome. The command 'findPeaks' from the HOMER tool package (http://homer.salk.edu/homer/) was used to identify enriched regions in the Bcl9 immunoprecipitation samples using the '-style = factor' option (routinely used for transcription factors with the aim of identifying the precise location of DNA-protein contact). Input and IgG samples were used as enrichment-normalization controls. Peak calling parameters were adjusted as following: L = 4 (filtering based on local signal), F = 4 (fold-change in target experiment over input control). Annotation of peaks' position (i.e. the association of individual peaks to nearby annotated genes) was obtained by the all-in-one program called 'annotatePeaks.pl'. Finally, the HOMER command 'makeUCSCfile' was used to produce bed-Graph formatted files that can be uploaded as custom tracks and visualized in the UCSC genome browser (http://genome.ucsc.edu/).

## Association analyses in the DiscovEHR cohort

Sample preparation and sequencing was performed as described previously. Exome capture was performed with either NimbleGen (SeqCap VCRome). Sequence reads were aligned to GRCh38. Variant calling and genotyping was performed as described in detail previously (*Dewey et al., 2016*). Gene definitions were restricted those genes and transcripts with annotated start and stop codons; a total of 19,467 protein-coding genes.

Genotypes were coded as follows: homozygous reference as 0, heterozygotes as 1, and homozygous alternative or compound heterozygous as 2. Only individuals of European ancestry were analyzed. For clinical diagnoses, ICD-10 codes were collapsed into hierarchical clinical disease groups and corresponding controls using a modified version of the groupings proposed previously (*Denny et al., 2010*; *Denny et al., 2013*). Case assignment for the 14,128 ICD-10 based diagnoses required one or more of the following: a problem list entry of the diagnosis code or an encounter diagnosis code entered for two separate clinical encounters on different calendar days. Firth logistic regression was performed using PLINKv1.9 (*Chang et al., 2015*), with adjustment for age, age squared, sex, and the first four principal components of ancestry. Gene-trait association results meeting the Bonferroni corrected significance threshold 0.05/268192 (p<1.86–7) are reported.

## Statistical analysis

Methods for statistical analysis are specific to various components of the study and are described in detail for each type of experiment.

## Acknowledgements

Funded by NIH-NIAMS R01 AR055904 (to VR). Micro-computed tomography and histology provided by Massachusetts General Hospital Skeletal Research Core (NIH P30 AR066261). We thank Aris Economides for sclerostin neutralizing antibody; The Geisinger-Regeneron DiscovEHR Collaboration for enabling phenome-association wide studies; the University of Iowa Gene Transfer Vector Core (supported in part by the NIH and the Roy J Career Foundation) for viral vectors. VS, LC, JN, CC, DZ, SP, KC, SO, MF and LG performed experiments and data analysis. VS and VR prepared the manuscript with input from co-authors. We have no competing interests.

## Additional information

### Competing interests

Nehal Gosalia, Aris Economides: Employee of Regeneron Pharmaceuticals. There are no other competing interests to declare. The other authors declare that no competing interests exist.

## Funding

| Funder | Grant reference number | Author |
| --- | --- | --- |
| National Institute of Arthritis and Musculoskeletal and Skin Diseases | R01 AR055904 | Vicki Rosen |

The funders had no role in study design, data collection and interpretation, or the decision to submit the work for publication.

## Author contributions

Valerie S Salazar, Conceptualization, Resources, Data curation, Formal analysis, Validation, Investigation, Visualization, Methodology, Writing—original draft, Writing—review and editing; Luciane P Capelo, Claudio Cantù, Data curation, Formal analysis, Investigation, Methodology, Writing—review and editing; Dario Zimmerli, Conceptualization, Formal analysis, Investigation, Methodology, Writing—review and editing; Nehal Gosalia, Conceptualization, Data curation, Formal analysis, Validation, Investigation, Visualization, Methodology, Writing—review and editing; Steven Pregizer, Karen Cox, Satoshi Ohte, Marina Feigenson, Data curation, Investigation, Methodology; Laura Gamer, Funding acquisition, Investigation, Methodology; Jeffry S Nyman, Formal analysis, Supervision, Investigation, Methodology, Writing—review and editing; David J Carey, Resources, Data curation, Formal analysis, Supervision, Project administration; Aris Economides, Resources, Supervision, Project administration; Konrad Basler, Resources, Supervision, Funding acquisition; Vicki Rosen, Conceptualization, Resources, Supervision, Funding acquisition, Writing—review and editing

## Author ORCIDs

Valerie S Salazar (iD) http://orcid.org/0000-0002-2111-9313
Aris Economides (iD) https://orcid.org/0000-0002-6508-8942
Vicki Rosen (iD) http://orcid.org/0000-0002-4029-1055

## Ethics

Animal experimentation: In vivo experiments were performed in compliance with the Guide for the Care and Use of Laboratory Animals and were approved by the Harvard Medical Area Institutional Animal Care and Use Committee (protocol #04043 to Vicki Rosen).

## Decision letter and Author response

Decision letter https://doi.org/10.7554/eLife.42386.032
Author response https://doi.org/10.7554/eLife.42386.033

# Additional files

## Supplementary files

• Transparent reporting form
DOI: https://doi.org/10.7554/eLife.42386.028

## Data availability

The DiscoverEHR human dataset is published and publicly available at http://www.discovehrshare.com. Using search terms 'BMP2' and 'GRHL3' and a Bonferroni significance threshold of $P < 1.86e-7$ for 268,192 association results, we observed three significant associations for BMP2 and six significant associations for GRHL3. Mouse limb bud ChIP-sequencing data are available through ArrayExpress website https://www.ebi.ac.uk/arrayexpress and accession #E-MTAB-7652.

The following dataset was generated:

| Author(s) | Year | Dataset title | Dataset URL | Database and Identifier |
| --- | --- | --- | --- | --- |
| Cantù C, Zimmerli D, Basler K | 2019 | Mouse Limb Bud Bcl-9 Chromatin Immunoprecipitation and Sequencing | https://www.ebi.ac.uk/arrayexpress/experiments/E-MTAB-7652/ | ArrayExpress, E-MTAB-7652 |

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
