## [Decision Letter]

Thank you for submitting your article "Reactivation of a Developmental *Bmp2* Signaling Center is Required for Therapeutic Control of the Periosteal Niche" for consideration by *eLife*. Your article has been reviewed by three peer reviewers, and the evaluation has been overseen by a Reviewing Editor and Harry Dietz as the Senior Editor. The following individual involved in review of your submission has agreed to reveal her identity: Karen Lyons (Reviewer #1). The other reviewers remain anonymous.

The reviewers have discussed the reviews with one another and the Reviewing Editor has drafted this decision to help you prepare a revised submission.

Overall there was significant enthusiasm for the work, which was felt to be novel and potentially important for the field, particularly in regards to the role of BMP2 downstream of Wnts in the periosteum. There were concerns which should be addressed however in the revision. There is a need to clarify the ChipSeq data; There was a question about role of other BMPs besides BMP2 in the periosteum, and more discussion about the notable finding that neither PTH nor anti SOST antibodies reversed the fracture phenotype of the cKOs; furthermore there remain some questions about the skeletal phenotype of the cKOs especially in the limb that will require an additional figure.

*Reviewer #1:*

This is a thorough characterization of the unique periosteal phenotype in *Bmp2* mutant mice. Using a combination of in vivo and ex vivo approaches, the authors show that the *Bmp2* mutant phenotype results from loss of local production of *Bmp2* expression in PRx1+ve progenitors. The histomorphometry showing increased osteocyte numbers in cortical bone suggests that in the absence of BMP2, progenitors differentiate prematurely. The data also show that the differentiation itself is impaired. These are important discoveries. So is the discovery of a BMP signaling center in the inner periosteum. This finding alone will attract high interest among bone biologists. Furthermore, the authors show that exogenous GDF5 and BMP7 cannot compensate for BMP2. This is a surprising and very important finding, hinting that relative affinities of ligands for their receptors is decisive for cell fate decisions by progenitors. A large amount of data is presented in this manuscript, encompassing phenotypic, transcriptome and ChIP-seq analysis, and human exome sequence data to document that BMP2 acts downstream of Wnts to activate expression of Sp7. The discussion is concise and the authors do an outstanding job distilling a great deal of experimental evidence to support their important conclusions.

*Reviewer #2:*

In this study, the authors have further examined the roles on BMP2 in postnatal periosteal bone growth and functioning. The data indicate that ablation of *Bmp2* in the Prx1-Cre lineage is sufficient to impair not only long bone width growth, but also fracture repair and response to anabolic bone therapies such as intermittent PTH or sclerostin antibody treatment. Additional data indicate that BMP2 acts downstream of LRP5/6 signaling since periosteal and fracture defects in *Bmp2*;Prx1-Cre mutants were not rescued by haplo-insufficiency of Dkk1 or treatment with SOST antibody. ChIP and deep sequencing data suggest that BMP2 affects expression of Sp7 (required for osteoblast differentiation) by recruitment of Smad1 and Grhl3 at conserved regulatory elements. Re-examination of DiscovEHR data sets pointed to links between BMP2 and GRHL3 variants and increased fracture risks in patients. The authors conclude that *Bmp2* expression in periosteal progenitors accounts for all the key postnatal functions of this tissue, including long bone width, fracture repair and anabolic responses.

This is an important and comprehensive study from a group who has contributed much and in an original manner to this area of biomedical research. For the most part, the data are strong and convincing and do strengthen the argument that BMP2 plays a rather unique and non-redundant role in establishing the phenotypic properties and repair capacity of periosteum. As the authors stress, future identification of means to stimulate periosteal BMP2 action or expression could lead to new therapies to strengthen or repair bone. There are only a few and relatively minor concerns.

A key conclusion reached is that "although many BMPs are expressed in bone, periosteal BMP signaling and bone formation require only *Bmp2* in the Prx1-Cre lineage". It would be important to know whether *Bmp2* ablation may have resulted in changes in expression of other Bmps, particularly in periosteum. Also, given that Prx1-Cre does not target only the periosteal progenitors but the entire mesenchymal cell population within the limbs, it would be important for the authors to discuss possible changes in tissue interactions within the mutant limbs influencing outcomes.

As the authors correctly underline, it is quite interesting that Dkk1 haploinsufficiency and treatment with iPTH or SOST antibody did not appear to rescue periosteal growth and fracture repair defects in the *Bmp2*;Prx1-Cre mice. In fact, the latter treatments increased the frequency of spontaneous fractures in mutant mice. The authors mention, but do not deal with, this striking outcome, and it would be important to not only face it, but also try to account for it.

The ChIP-sequencing data in Figure 6 are somewhat perplexing as they were generated using whole E10.5 mouse limb buds and immortalized E13.5 limb bud cells in vitro. Both of these represent early embryonic and highly heterogenous cell populations, and it is hard to know to what extent they speak (or can speak) about the function of selected genes (Axin2, Lef1 and Grhl3) specifically in periosteal cells. The fact that *Bmp2* ablation does not affect embryonic long bone development and that bone formation has not started by E13.5 also question the significance of the data.

*Reviewer #3:*

This manuscript from Salazar and colleagues presents information of potential interest to many in the field. While the concepts being addressed are of interest, there are numerous points that should be clarified that are listed below.

1) Please show the bone phenotypes (Figure 1) for Prx1-*Bmp2* heterozygotes if they are available. Please show a schematic diagram or a reference on *Bmp2*-Lacz allele (Figure 2A).

2) Although LacZ staining and GFP fluorescence (Figure 2) show *Bmp2* expression and *Bmp2* activation nicely, sometimes the signal could be too weak to detect due to the limitation of the techniques. Such as there is a significant *Bmp2* activation (GFP, Figure 1U) in the adult bone marrow, but there is no *Bmp2* expression (LacZ, Figure 1H and I) at the same place. So, a qPCR with different bone fractions by serial collagenase digestion of cortical bone would be a good confirmation on the location of Bmp2 in different stages.

3) No Col1a1 expression in WT/cKO osteocytes within the cortical bone (Figure 3F)?

4) *Bmp2*-Prx1 cKO appears to not affect iPTH-induced trabecular bone gain (Figure G and N). The author claims that *Bmp2* is a mediator of iPTH for periosteal bone formation. However, there was no significant change in periosteal expansion in either WT or cKO mice (Figure 4H). It is also not clear why there was a significant increase in periosteal bone formation in the cKO mice as well (Figure 4J and L).

5) Figure 4M is a little confusing. According to Figure 2H, *Bmp2* is expressed in adult bone osteocytes (not periosteal cells). However, iPTH only induces ID3 in only periosteal cells but not osteocytes themselves. Is this suggesting that osteocytes do not respond to *Bmp2* although they express *Bmp2*? The reviewer is also confused by the cKO bone morphology in Figure 4M, which shows massive woven bones and relative stronger (compared with the WT) ID3 expression in osteocytes.

6) Figure 4Q-T do not have error bars or statistical analysis done.

7) What does no Imin change in *Bmp2* cKO mice mean in Figure 5I?

8) It is convincing that *Bmp2* cKO inhibits anti-SOST-induced bone expansion (Figure 5G-J). Since there is no bone phenotype when *Bmp2* is deleted in osteoblasts (Sp7-cre or Col1-cre) and BV/TV is normal in Prx1-*Bmp2* cKO mice (Figure 4G and N), it appears that Bmp2 is important for periosteum, but not for trabecular bone formation. What is the hypothesis that BV/TV cKO is not increased by anti-SOST (Figure 5F) other than development versus treatment?

9) Please provide an additional and higher powered picture for Figure 5O. It seems that all Prx1+ cells are BRE/GFP+ while all Prx1- cells are BRE/GFP-.

10) Also, please provide an additional picture without DAPI/blue fluorescence for Figure 5N to show GFP in the periosteum.

11) Compare the result of Bcl9 ChIP-seq in limbs with publicly available Tcf4/β-catenin Chipseq data in other tissues/cell lines in terms of pathway enrichments. Is there a reason to choose Bcl9 antibody over Tcf4 antibody?

12) Bmp2 is proposed to be a direct downstream target of canonical Wnt signaling (Figure 6E) in limb buds. Does Wnt3a treatment directly induce Bmp2 mRNA in limb bud cells?

13) Are the authors proposing that Bmp2-Sp7-Col1a1 cascade is only present in the periosteum, but not in cortical bone or trabecular bone?

---

## [Author Response]

Reviewer #2:

[…] This is an important and comprehensive study from a group who has contributed much and in an original manner to this area of biomedical research. For the most part, the data are strong and convincing and do strengthen the argument that BMP2 plays a rather unique and non-redundant role in establishing the phenotypic properties and repair capacity of periosteum. As the authors stress, future identification of means to stimulate periosteal BMP2 action or expression could lead to new therapies to strengthen or repair bone. There are only a few and relatively minor concerns.A key conclusion reached is that "although many BMPs are expressed in bone, periosteal BMP signaling and bone formation require only Bmp2 in the Prx1-Cre lineage". It would be important to know whether Bmp2 ablation may have resulted in changes in expression of other Bmps, particularly in periosteum.

New data in Figures 10A-B show that *Bmp7* is also expressed in the periosteum and therefore does not compensate for loss of *Bmp2*. This is fully consistent with the in vitro data showing that recombinant BMP7 does not compensate for loss of *Bmp2* in primary osteoblast cultures. Expression of *Bmp4* in periosteal cells was at the lower limit of detection by QPCR and therefore also cannot compensate for lack of *Bmp2*.

Also, given that Prx1-Cre does not target only the periosteal progenitors but the entire mesenchymal cell population within the limbs, it would be important for the authors to discuss possible changes in tissue interactions within the mutant limbs influencing outcomes.

Based on our *Bmp2-lacZ data*, we do not see evidence of *Bmp2* expression that would influence other tissue interactions within the mutant limbs and that it is likely that if BMP signaling is required for those interactions, other BMPs present in the limb would provide the signal. While it is somewhat surprising that BMP2 would have such a highly specific role in limb skeletogenesis, the role is fully consistent with the *Bmp2-lacZ* localization pattern. As genetic ablation of *Bmp4* or *Bmp7* do not affect periosteal apposition but have other ligand-specific effects on the limb skeleton (2008 Tsuji, 2010 Tsuji, and 2006 Bandyopadhyay), we believe ligand localization is a major determinant of this phenotype.

As the authors correctly underline, it is quite interesting that Dkk1 haploinsufficiency and treatment with iPTH or SOST antibody did not appear to rescue periosteal growth and fracture repair defects in the Bmp2;Prx1-Cre mice. In fact, the latter treatments increased the frequency of spontaneous fractures in mutant mice. The authors mention, but do not deal with, this striking outcome, and it would be important to not only face it, but also try to account for it.

We observed increased frequency of spontaneous fractures only with iPTH treatment, as shown in Figures 2-T. It is well established that iPTH activates bone resorption alongside bone formation, where the net effect in wildtype mice is increased bone mass. In the absence of BMP2, activation of bone formation response is severely blunted and thus the altered balance shifts towards increased bone fragility.

The ChIP-sequencing data in Figure 6 are somewhat perplexing as they were generated using whole E10.5 mouse limb buds and immortalized E13.5 limb bud cells in vitro. Both of these represent early embryonic and highly heterogenous cell populations, and it is hard to know to what extent they speak (or can speak) about the function of selected genes (Axin2, Lef1 and Grhl3) specifically in periosteal cells. The fact that Bmp2 ablation does not affect embryonic long bone development and that bone formation has not started by E13.5 also question the significance of the data.

Question 1) Heterogeneity

- For E10.5, the ChIP-seq analysis automatically enriches spefically for cells that are actively engaged in canonical Wnt signaling.

- For E13.5, the immortalized limb bud cells we used are a clonal line of osteochondral-progenitors, not a heterogeneous population (Rosen, 1994).

Question 2) Time points were selected for analysis

- E13.5 is when Osx1+ osteoblasts and the bone collar (primitive periosteum) first appear (Rodda and McMahon, 2006). These cells persist for the first 30 days of life and directly mediate periosteal bone growth during early post-natal life (Maes, 2010).

Question 3) Function of selected genes

- *Axin2* and *Lef1* are only used as positive controls to demonstrate that the Bcl9-pulldown method enriches for previously established canonical Wnt target genes.

- We propose that Grhl3 acts as a co-factor with Smads for induction of *Sp7*. For this to be true, *Grhl3* would need to be induced prior to induction of *Sp7*, and this hypothesis is confirmed in this figure.

Reviewer #3:

This manuscript from Salazar and colleagues presents information of potential interest to many in the field. While the concepts being addressed are of interest, there are numerous points that should be clarified that are listed below.1) Please show the bone phenotypes (Figure 1) for Prx1-Bmp2 heterozygotes if they are available. Please show a schematic diagram or a reference on Bmp2-Lacz allele (Figure 2A).

The phenotype of *Bmp2; Prx1-Cre* heterozygotes has been previously shown in Tsuji et al., 2006.

A schematic showing the structure of the *Bmp2-lacZ* allele has been provided in new Figure 2—figure supplement 1A-C.

2) Although LacZ staining and GFP fluorescence (Figure 2) show Bmp2 expression and Bmp2 activation nicely, sometimes the signal could be too weak to detect due to the limitation of the techniques. Such as there is a significant Bmp2 activation (GFP, Figure 1U) in the adult bone marrow, but there is no Bmp2 expression (LacZ, Figure 1H and I) at the same place. So, a qPCR with different bone fractions by serial collagenase digestion of cortical bone would be a good confirmation on the location of Bmp2 in different stages.

We thank the reviewer for the comments and apologize for not being clear about how the signaling reporter works. The *BRE-gfp* reporter is not specific for BMP2 signaling but rather provides a net readout of all BMP signaling. The observation that GFP expression is maintained in the marrow compartment of *Bmp2* cKOs is consistent with the fact that many BMPs besides *Bmp2* have been shown previously by several groups to be expressed in the bone marrow where they contribute to hematopoiesis as well as skeletal biology. By the same rationale, the observation that GFP persists in the marrow of *Bmp2* cKOs while being attenuated in the periosteum, strongly supports our model that other BMPs do not compensate for BMP2 in the periosteum. We propose that BMP2 is the main BMP family member with activity in the periosteum, whereas other BMPs, such as *Bmp4* and *Bmp7* compensate in other compartments (2008 Tsuji, 2010 Tsuji, and 2006 Bandyopadhyay).

3) No Col1a1 expression in WT/cKO osteocytes within the cortical bone (Figure 3F)?

In a lineage-tracing experiment with *Col1a1-Cre*, osteocytes would be indeed be “marked” as descendants of a population that at some point expressed bona fide *Col1a1* mRNA. However, the primary function of bone-embedded osteocytes is no longer matrix-production but mechanical sensing. As part of this change in cell function, osteocytes would not necessarily be expected to be actively transcribing and depositing new *Col1a1* in the surrounding bone as unabated production of collagen would eventually eliminate the lacunar space they occupy. The observation that osteocytes are not transcribing *Col1a1* mRNA, regardless of *Bmp2* expression, is therefore the expected outcome.

4) Bmp2-Prx1 cKO appears to not affect iPTH-induced trabecular bone gain (Figure G and N). The author claims that Bmp2 is a mediator of iPTH for periosteal bone formation. However, there was no significant change in periosteal expansion in either WT or cKO mice (Figure 4H). It is also not clear why there was a significant increase in periosteal bone formation in the cKO mice as well (Figure 4J and L).

Periosteal BFR and MAR are direct readouts of periosteal function while uCT total area is indirect. Increased total area requires a longer amount of time to be measurable compared to increased BFR/MAR since it is a cumulative effect over time of bone forming activity.

5) Figure 4M is a little confusing. According to Figure 2H, Bmp2 is expressed in adult bone osteocytes (not periosteal cells). However, iPTH only induces ID3 in only periosteal cells but not osteocytes themselves. Is this suggesting that osteocytes do not respond to Bmp2 although they express Bmp2? The reviewer is also confused by the cKO bone morphology in Figure 4M, which shows massive woven bones and relative stronger (compared with the WT) ID3 expression in osteocytes.

Figure 2H shows *Bmp2-lacZ* expression in osteocytes of untreated juvenile mice of 2 weeks of age. These data cannot be directly compared to histology in 4M which shows the cortical bone of a 1 month old animal that was treated for the past 2 weeks with iPTH.

The authors thank the reviewer for pointing out the highly porous nature of the PTH-treated cKO cortical bone in Figure 4M. We have used the lamellar structures as guides to distinguish true cortical bone with osteocytes from the intra-cortical pockets of heterogeneous endosteum and marrow. This greatly clarifies that Id3+ nuclei can be found in the periosteum and marrow of PTH-treated WT mice, but not the periosteum of PTH-treated cKO mice. Importantly, osteocytes in cortical bone of PTH-treated mice are Id3-, regardless of *Bmp2* genotype. The data across the paper are therefore consistent in that osteocytes are negative for Id1, Id3, and *Col1a1*, as would be expected for cells that should not be actively engaged in BMP signaling and are not making bone matrix.

6) Figure 4Q-T do not have error bars or statistical analysis done.

Data in Figure 4Q represent an absolute number (% of total animals scored with a binary outcome). Error bars and statistical analysis are therefore not applicable here.

7) What does no Imin change in Bmp2 cKO mice mean in Figure 5I?

Imin is the minimum value of moment of inertia for the bone, which occurs relative to the cross-sectional orientation with the least resistance to bending. Wildtype mice treated with sclerostin antibody exhibit increased Imin while cKO mice exhibit no change. This estimated material property is based on diaphyseal (cortical) morphology obtained by microCT, and predicts that wildtype bones, but not cKO bones are more resistant to bending (and fracture) after treatment.

8) It is convincing that Bmp2 cKO inhibits anti-SOST-induced bone expansion (Figure 5G-J). Since there is no bone phenotype when Bmp2 is deleted in osteoblasts (Sp7-cre or Col1-cre) and BV/TV is normal in Prx1-Bmp2 cKO mice (Fig4G and N), it appears that Bmp2 is important for periosteum, but not for trabecular bone formation. What is the hypothesis that BV/TV cKO is not increased by anti-SOST (Figure 5F) other than development versus treatment?

In Figure 10C, in vitro experiments reveal that BMSC cells exhibit a “bell-shaped curve” response to ectopically activated WNT signaling, where low concentrations of recombinant Wnt3a activate matrix calcification but higher amounts become inhibitory. This inhibitory WNT threshold is lowered in the absence of *Bmp2*, as BMSC lacking *Bmp2* are able to mineralize under normal conditions but not when exposed to recombinant Wnt3a. In other words, the inhibitory WNT threshold is reduced when the net amount of BMP signaling is reduced. in vivo, we hypothesize that endosteal osteoblasts are able to function in an unchallenged environment due to compensation by other BMPs (probably BMP4/7), but that the reduced net strength of pan-BMP signaling has made them more sensitive to Wnt-inhibition. In a future study, this interesting possibility could be measured by monitoring bone mass in *Bmp2; Prx1-Cre* mice treated with decreasing doses of SOST-ab.

9) Please provide an additional and higher powered picture for Figure 5O. It seems that all Prx1+ cells are BRE/GFP+ while all Prx1- cells are BRE/GFP-.

A new supplementary figure with separate images for GFP, DAPI, and RFP channels very clearly shows that the vast majority of cells derived from the *Prx1-Cre; TdTomato* lineage are not actively engaged in BMP signaling (GFP+).

10) Also, please provide an additional picture without DAPI/blue fluorescence for Figure 5N to show GFP in the periosteum.

A new figure panel has been added to Figure 5P.

11) Compare the result of Bcl9 ChIP-seq in limbs with publicly available Tcf4/β-catenin Chipseq data in other tissues/cell lines in terms of pathway enrichments. Is there a reason to choose Bcl9 antibody over Tcf4 antibody?

TCF/LEF proteins are constitutively bound to chromatin. In a WNT-OFF state, they function as transcriptional repressors of their bound target genes. In a WNT-ON state, additional co-factors (such as β-catenin, pygopus, and Bcl9) are recruited to genome-bound TCF, thereby converting TCFs from transcriptional repressors to transcriptional activators. TCF pulldown is therefore not able to discriminate between genes that are being repressed or induced, and thus a co-activator must be used to understand the subset of target genes that are being activated at any given time.

We used Bcl9 pulldown to selectively enrich for loci occupied by transcriptionally active β-catenin (by definition, only in WNT-ON cells). In support of our scientific rationale, Bcl9 mutant mice exhibit defects in distal limb and long bone development, demonstrating at the genetic level that β-catenin transcription in the developing limb is fully dependent on the Bcl9-β-catenin interaction (2018 Cantu Gen Dev).

12) Bmp2 is proposed to be a direct downstream target of canonical Wnt signaling (Figure 6E) in limb buds. Does Wnt3a treatment directly induce Bmp2 mRNA in limb bud cells?

Yes, a new Figure 11—figure supplement 1A shows that *Bmp2* mRNA is upregulated by recombinant Wnt3a.

13) Are the authors proposing that Bmp2-Sp7-Col1a1 cascade is only present in the periosteum, but not in cortical bone or trabecular bone?

We propose that a general BMP-dependent cascade acts upstream of the *Sp7/Col1a1*. In the periosteum, our data provide evidence that this BMP-dependent cascade is activated primarily if not entirely by BMP2. Our study does not address which BMPs compensate for loss of BMP2 at non-periosteal sites. However, it reasonable to hypothesize that the major compensatory BMP at non-periosteal sites is BMP4, since mice with compound loss of *Bmp2* and *Bmp4* in *Prx1-Cre* lineage fail to make an appendicular skeleton (2006 Bandyopadhyay) while mice with loss of only *Bmp2* manifest a periosteal-specific phenotype.